# Alternatively spliced *NFKB1* transcripts enriched in Andean Aymara modulate inflammation, HIF and hemoglobin

Jihyun Song [1,8], Seonggyun Han [2,3,8], Ricardo Amaru[4,8], Lucie Lanikova [5], Teddy Quispe[4], Dongwook Kim[2], Jacob E. Crawford[6], Soo Jin Kim[1], Younghee Lee [7] ✉ & Josef T. Prchal [1] ✉

The molecular basis of increased hemoglobin in Andean Aymara highlanders is unknown. We conducted an integrative analysis of whole-genome-sequencing and granulocytes transcriptomics from Aymara and Europeans in Bolivia to explore genetic basis of the Aymara high hemoglobin. Differentially expressed and spliced genes in Aymaras were associated with inflammatory and hypoxia-related pathways. We identified transcripts with 4th or 5th exon skipping of *NFKB1* (AS-*NFKB1*), key part of NF-kB complex, and their splicing quantitative trait loci; these were increased in Aymaras. AS-*NFKB1* transcripts correlated with both transcripts and protein levels of inflammatory and HIF-regulated genes, including hemoglobin. While overexpression of the AS-*NFKB1* variant led to increased expression of inflammatory and HIF-targeted genes; under inflammatory stress, NF-kB protein translocation to the nucleus was attenuated, resulting in reduced expression of these genes. Our study reveals AS-*NFKB1* splicing events correlating with increased hemoglobin in Aymara and their possible protective mechanisms against excessive inflammation.

There are three widely studied populations living at high altitudes: Tibetans, Ethiopians, and Andeans. The molecular mechanisms related to oxygen sensing are the foundation of evolutionary adaptations in these populations to the hypoxic high-altitude environment[1]. Regulation of erythropoiesis plays significant roles in adaptations for oxygen-sensing and tissue oxygen delivery[2]. High hemoglobin (Hb) levels in a hypoxic environment offer the potential benefit of increased oxygen retention in the human body. However, a potentially harmful effect of increased blood viscosity from high hematocrit (Hct) also exists and may contribute to chronic mountain sickness[3].

It was assumed that some high-altitude populations, such as Tibetans and Ethiopians, have undergone natural selection that blunted the normal physiological response maintaining of elevated Hb levels at high altitude[1]. In Tibetans, two genetic variants in the hypoxia sensing pathway, regulated by hypoxia-inducible factors (HIFs), that regulate erythropoiesis, have undergone positive selection to prevent high Hb and Hct. These are gain-of-function variants in the principal HIFs negative regulator *EGLN1* (encoding prolyl hydroxylase 2; PHD2), and a Denisovan-originated haplotype of the *EPAS1* gene[4], that encodes the principal regulator of erythropoietin (EPO), HIF-2α. However, as previously reported, Tibetan Sherpas maintain normal Hb concentration at high altitudes, but exhibit increases in both plasma and red cell volumes, i.e., *true erythrocytosis* that is masked by increased plasma volume. This combination is advantageous for delivering oxygen to tissues due to

[1]Division of Hematology and Hematologic Malignancies, Huntsman Cancer Institute, University of Utah and VA Hospital, Salt Lake City, UT, USA. [2]Department of Biomedical Informatics, School of Medicine, University of Utah, Salt Lake City, UT, USA. [3]Department of Psychiatry, School of Medicine, University of Utah, Salt Lake City, UT, USA. [4]Cell Biology Unit, School of Medicine, San Andres University, National Academy of Sciences, La Paz, Bolivia. [5]Department of Cell and Developmental Biology, Institute of Molecular Genetics of the Czech Academy of Sciences, Prague, Czech Republic. [6]Verily Life Sciences, South San Francisco, CA, USA. [7]College of Veterinary Medicine and Research Institute for Veterinary Science, Seoul National University, Seoul, Republic of Korea. [8]These authors contributed equally: Jihyun Song, Seonggyun Han, Ricardo Amaru. ✉e-mail: amazon@snu.ac.kr; josef.prchal@hsc.utah.edu

extended capillary circulation[5], while avoiding the potential harmful effects of high Hct and blood viscosity. In contrast to Tibetans and other high-altitude adapted populations, Andeans have distinct adaptations for Hb phenotypes. The Andeans have Hb levels that are not only higher than either Tibetans or Ethiopians[1] but even higher than European sojourners to high altitude[6]. In contrast to Tibetans, Andeans' higher red cell volume is not accompanied by a beneficial expansion of plasma volume, resulting in their elevated Hb levels[7]. This leads to high blood oxygen saturation[6,7]. Andeans' erythroid progenitors were reported to be hypersensitive or even to grow without EPO in vitro, suggesting that they also have some features of primary erythrocytosis[8]. However, the molecular mechanisms responsible for the high Hb observed in Andeans have remained elusive. We hypothesized that the elucidation of these unique Andean molecular mechanisms would enhance our understanding of human hypoxia adaptations.

The Aymara and Quechuas are two distantly related but distinct and well-studied Andean indigenous highland populations[9]. We previously reported genomic signatures of natural selection in Aymara[10] that differ in some respects from those of Quechuas[9]. In this study, we focus solely on Aymara high altitude adaptation and their Hb levels.

Erythron production is controlled by HIFs to assure optimal tissue oxygen delivery, with HIF-2 being the principal regulator of *EPO* transcription[1,11]. Whole-genome-sequencing (WGS) analysis of high altitude populations including Tibetans[12–14], Ethiopians[1], and Quechuas[15] have identified signals of natural selection: strong single nucleotide polymorphisms (SNPs) selection signals in *EPAS1* and *ELGN1* in Tibetans and *SENP1* and *ANP32D* genes in Quechuas. These genes except *ANP23D* are all involved in the HIF pathway and are associated with regulation of Hb at high altitudes. However, these genes were not found to be evolutionary selected in Andean Aymara[16]. Our previous WGS study failed to discover the mechanisms of high Hb levels in Aymara; however, we found strong selection signals of *BRINP3*, *NOS2*, and *TBX5*, which are related to cardiovascular function and development[10]. While *BRINP3* also regulates inflammation by modulating NF-kB activity in humans[17], *BRINP3* haplotype did not correlate with Aymara high Hb[16].

Transcriptomic changes, i.e., alternatively regulated genes: decreased or increased transcripts, or multiple transcript isoforms, may also play a role in response to the hypoxic environment. The *CASK* gene has differential expression and alternative splicing forms (i.e., exon skipping or inclusion) under hypoxia and was reported to be enriched in hypoxia-regulated angiogenesis[18–21]. We hypothesized that the integration of studies of evolutionary selected genetic variants (i.e., SNPs) analysis with concomitant analysis of transcriptional regulation at the splicing level could further expand our understanding of evolutionary adaptation to hypoxia. We applied quantitative trait locus analysis of genes and spliced exons to develop a more precise estimation of these two genetic markers, SNPs, and gene expression in an integrative framework. We postulated this approach might also identify previously undescribed gene expression and splicing variants and provide new insights into the mechanism underlying the distinct Aymaras' high Hb levels at high altitude.

As transcriptomes are cell lineage- and differentiation-specific, we attempted to isolate pure cell lineages from small amount of available peripheral blood but were only able to obtain high-quality granulocytes' RNA that we used for whole-transcriptome analysis. Granulocytes release various pro-inflammatory cytokines and chemokines to initiate the inflammatory response; these inflammatory changes may also influence erythropoiesis. We performed an integrative analysis of granulocytes transcriptomes to prioritize candidate genes linked to high Hb in Aymara.

## Results

### Differentially expressed genes in Aymara
The overall design of the computational analysis in this study is described in Supplementary Fig. S1. We first compared transcript expression levels from the unbiased granulocyte transcriptomes of 10 Aymara and 4 European controls living in La Paz (3650 m) and later confirmed these pilot data in a larger number of subjects (67 samples of Bolivian Aymara and 18 Europeans; see "Materials and Methods") to identify differentially expressed genes (DEGs). We found 1935 genes upregulated and 666 downregulated in Aymara compared to the European controls (adjusted $p$ value [Adj. $p$] < 0.05, |fold change| > 1.5; Fig. 1a and Supplementary Data 1). Gene set enrichment analysis identified these dysregulated genes as enriched in hematopoiesis, immune-related, and inflammatory-related Gene Ontology terms along with canonical pathways including T helper (Th17, Th1, and Th2) cell differentiation, T-cell receptor signaling pathway, human T-cell leukemia virus 1 infection, leukocyte activation, interleukin pathways, regulation of cytokine, and NF-κB related signaling pathway (Fig. 1d and Supplementary Data 2).

### Differential alternative splicing of genes in Aymara
We further analyzed the transcriptomic data to identify differential alternatively spliced genes (DSGs). This yielded a total of 1922 genes with alternative splicing events, comprising 1099 exon skipping events, 409 intron retention events, 161 alternative 3' splice sites, 136 alternative 5' splice sites, and 663 mutually exclusive exons (Fig. 1b, Supplementary Data 3). These alternatively spliced genes were also significantly over-represented in hematopoietic progenitor cell differentiation, HIF-1 signaling, immune-related, and inflammatory-related pathways including CD4 T cell receptor signaling, leukocyte activation, interleukin pathways, regulation of cytokine, TNF receptor, and NF-κB related signaling (Fig. 1d and Supplementary Data 4).

A heatmap with hierarchy clustering showed stratification of the Aymara and European samples based on DEG expression and percent spliced in (PSI) exon usage (Fig. 1c). Only a small portion of DEGs (10.2%) and DSGs (13.8%) overlapped at the gene level (upper Venn diagram in Fig. 1e), while at the pathway analysis multiple immune-related and NF-κB signaling pathways were shared by the two gene sets (Fig. 1d). Thus, although many individual genes may be exclusively regulated via expression or splicing mechanisms, the Aymara's differentially regulated genes may be possibly involved in those pathways that contribute to Aymara high-altitude adaptation.

### *Cis*-genetic regulators for DEGs and DSGs in Aymara
Since genetic variants (i.e., haplotypes and single nucleotide polymorphisms; SNPs) could act as regulators of DEGs and DSGs, we explored the SNPs associated with each individual DEG and DSG through integrative analysis of paired RNA-seq and WGS data. We considered SNPs within promoter regions as potential expression quantitative trait loci (eQTLs) for DEGs and those within alternatively spliced (AS) exons and their flanking introns as potential splicing quantitative trait loci (sQTLs) for DSGs (see Materials and Methods). Of the set of 2601 DEGs, a total of 160 SNPs were found to act as eQTLs associated with expression of 129 genes (Supplementary Data 5). Of the 1922 DSGs, a total of 858 SNPs were identified as sQTLs associated with altered exon usage of 340 genes (Supplementary Data 6). Only 7 genes were regulated by both eQTLs and sQTLs (bottom Venn diagram in Fig. 1e). Additionally, 57% of eQTL and 34% of sQTLs had a score less than or equal to 4 in RegulomeDB[22], which suggests they are in regulatory regions (Fig. 1f). Furthermore, among eQTLs and sQTLs, 37 eQTLs and 309 sQTLs were detected as Aymara adapted SNPs that showed differential allele frequencies compared to the general European populations (see "Methods") (Supplementary Data 7).

### *NFKB1* as a candidate hub gene in Aymara adaptation
We constructed a protein-protein interaction (PPI) network using the human phenotype resource in StringDB[23] for the 129 DEGs and 340 DSGs, respectively regulated by eQTLs and sQTLs to further gain insight into the molecular signature of these genes. The constructed

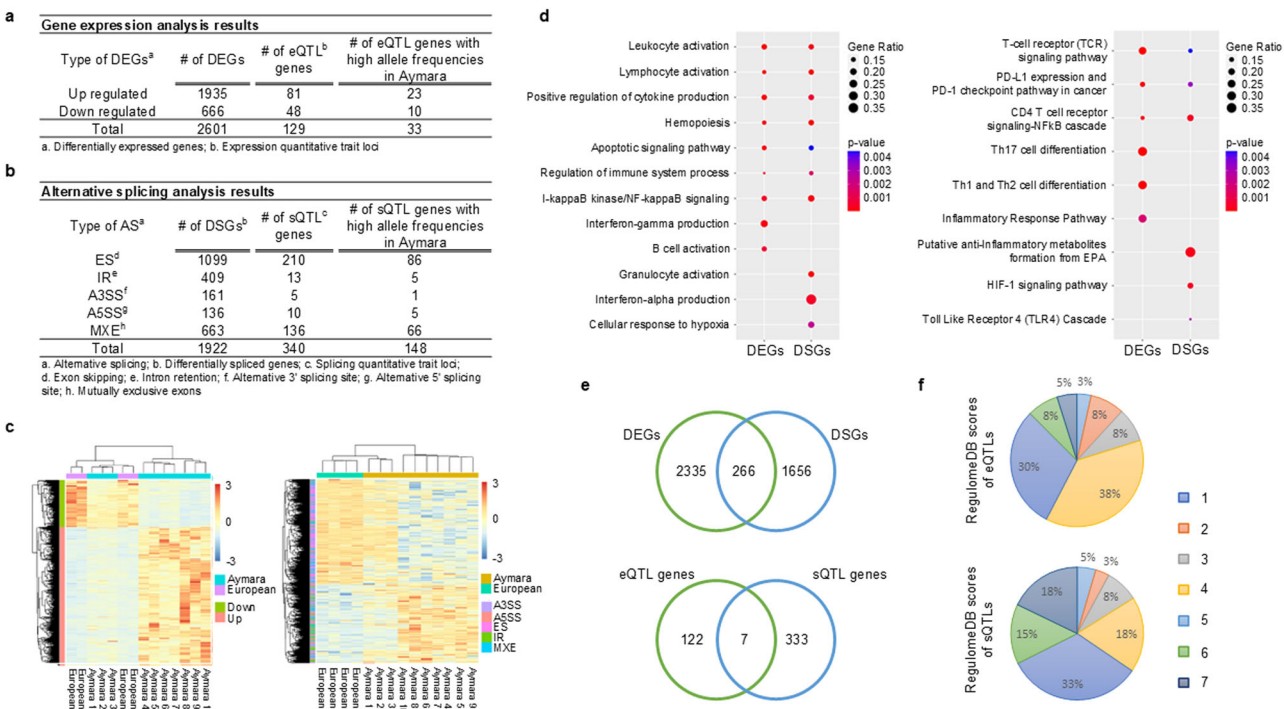

**Fig. 1 | Results of integrative analysis of RNA-seq and whole-genome sequencing (WGS) data. a** Identified differentially expressed genes (DEGs) and their expression quantitative trait loci (eQTLs). **b** Identified differentially alternatively spliced genes (DSG) and their splicing quantitative trait loci (sQTLs). **c** Heatmaps of DEG expression (left) and DSG exon usage (precent splicing in; PSI) (right). **d** Gene

Ontology terms and canonical pathways enriched in DEGs (left) and DSGs (right). *P* values were calculated using a one-sided Fisher's exact test. **e** Venn diagram of DEGs and DSGs (top), and of eQTL DEGs and sQTL DSGs (bottom). **f** RegulomeDB scores of eQTL regions (top) and sQTL regions (bottom).

network suggested *NFKB1* to be a hub gene; this gene has established roles in erythrocyte production, granulocyte count, and immune responses (Supplementary Fig. S2).

We detected previously unreported three alternative splicing of *NFKB1* (AS-*NFKB1*) in our RNA-seq analysis skipping either exon 4 (*NFKB1-AS1*), exon 5 (*NFKB1-AS2*), or both (*NFKB1-AS3*) (Supplementary Fig. S3a). Exons 4 and 5 together encode the Rel homology domain (RHD), DNA binding domain found in Pfam database[24], and therefore, all three exon skipping events may lead to the loss-of-function NFKB1 RHD domain. In addition, *NFKB1-AS1* and *NFKB1-AS3* encode peptides out-of-frame *NFKB1* mRNAs that would be expected to generate truncated, likely nonfunctional NF-κB proteins (Supplementary Fig. S3a). These are predicted to be destroyed by the nonsense-mediated decay (NMD) mechanism since they all have premature termination codons located more than 55 nucleotides upstream of the last exon of *NFKB1*[25]. As shown in Supplementary Fig. S3b, both (i.e., *NFKB1-AS1* and *NFKB1-AS3*) exon skipping events were more frequently observed in the Aymara population compared to Europeans: although *NFKB1-AS2* was detected in RNA-seq data, it was not significantly differentiated in Aymara since there were only a small number of the junction reads (data not shown). The integrative analysis of these Aymara transcriptomes and WGS data identified 11 SNPs that serve as sQTLs associated with skipping of exon 4 or 5 of *NFKB1* and have alternative allele frequencies higher in Aymara samples compared to European samples (Supplementary Data 7). NF-kB signaling plays an important role in erythropoiesis, inflammation, and the HIF-1 pathway. NFKB1 is a transcription factor that contributes to NF-kB level and activity. We, therefore, hypothesized that these sQTLs might be important for Aymara high-altitude adaptations (i.e., Hb level and inflammation) due to their affecting exon 4 and 5 splicing patterns in *NFKB1*. To explore this hypothesis, we further conducted a comprehensive functional analysis of *NFKB1* exon splicing (more frequent with

the sQTLs in Aymara), linking it to inflammation, Hb levels, and HIF-1 and HIF-2 pathways.

The detected sQTLs could be categorized into three groups: those with consistent genotypes among the ten Aymara samples (Supplementary Fig. S3c): (1) rs230525, rs230527, rs230517, and rs230516; (2) rs230519; and (3) rs230509, rs230511, rs230504, rs230491, rs230495, and rs230493. We selected representative SNPs for each group: two in group 1 (rs230525 and rs230527), one in group 2 (rs230519), and two in group 3 (rs230511 and rs230504) to further experimentally validate our *NFKB1* results. Associations of these five SNPs with AS-*NFKB1* and RNA-seq and WGS data are depicted in Supplementary Fig. S3d. For example, two SNPs in high linkage disequilibrium (LD) (Supplementary Fig. S3e) were in the intron between exon 4 and 5; an additional three SNPs were located between exon 5 and 6 (Supplementary Fig. S3a).

**Validation of *NFKB1* exon skipping events in an extended sample**

To validate the exon skipping events observed in our initial RNA-seq analysis, we analyzed larger number of Aymaras (*n* = 55), and Europeans (*n* = 18) living in La Paz (3650 m) and measured and compared expression levels of *NFKB1-AS1, NFKB1-AS2, and NFKB1-AS3* by qRT-PCR. Canonical *NFKB1* transcript expression was not significantly different, while transcript levels of *NFKB1-AS1*, *NFKB1-AS2*, and *NFKB1-AS3* were significantly higher in Aymara than in Europeans (Fig. 2a). In addition, we genotyped the five sQTLs in these extended samples and found a significant association with these AS-*NFKB1*, but not with canonical *NFKB1* transcript levels (Fig. 2b–e).

**Genetic variants of sQTLs enriched in the Aymara population**

Among the five sQTL SNPs, two shared the highest allele frequencies (AFs): rs230511-*T* and rs230504-*T*, with AF = 0.878 (Fig. 3a, b). These two SNPs belong to an LD block[26], and we selected rs230511 for further analysis.

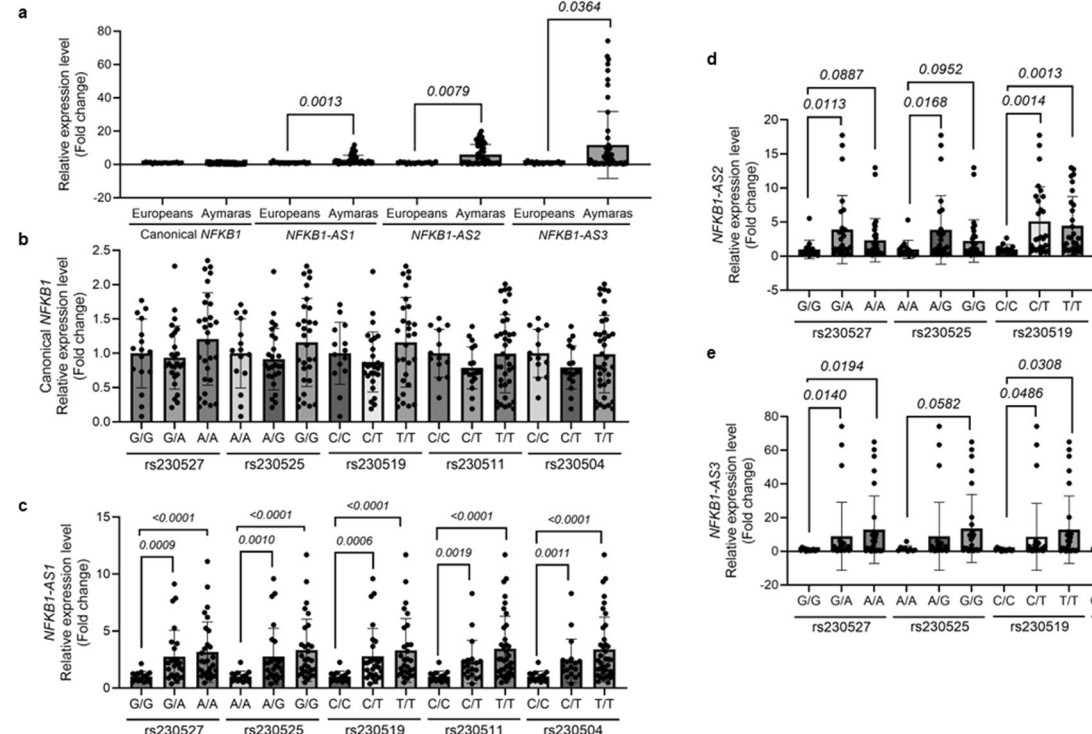

**Fig. 2 | Association of Aymara enriched *NFKB1* SNPs with alternatively spliced *NFKB1* transcripts. a** Canonical *NFKB1*, *NFKB1-AS1* (Exon 4 skipped AS-*NFKB1*), *NFKB1-AS2* (Exon 5 skipped AS-*NFKB1*), *NFKB1-AS3* (Exon 4 and 5 skipped AS-*NFKB1*) transcript levels in granulocytes of Aymara (*n* = 55) and European (*n* = 18) were measured and expressed as fold changes. **b** Canonical *NFKB1*, **c** *NFKB1-AS1*, **d** *NFKB1-AS2*, **e** *NFKB1-AS3* transcript levels of genotypes of rs230527, rs230525, 230519, rs230511, rs230504 were measured. *P* values were calculated by two-tailed Mann–Whitney test. All the data shown here are mean ± standard deviation (SD).

The allele rs230511-*T* was observed to be enriched in Aymara compared to the 1000 genome populations[27] (Fig. 3c and geographic distribution visualized[28] in Fig. 3d). The *T* allele was also more common in Aymara (AF = 0.878) than Tibetans (*n* = 72), another population living at high altitude; the frequency in Tibetans (AF = 0.364) was similar to that in Europeans (AF = 0.334, Fig. 3c).

We next collected and genotyped 188 samples of Bolivian Aymara living at different altitudes (*n* = 43 from Santa Cruz at 400 m, *n* = 133 from La Paz at 3650 m, and *n* = 12 from Chorolque at 5000 m), and found that the *T* allele frequency increased with altitude (Fig. 3f). We also observed that the *T* allele was more common in Aymara (*n* = 12) from Chorolque than in Quechuas (*n* = 49) at the same location (*p* = 0.00427, Fig. 3g); Quechuas had higher allele frequencies than Europeans, which is compatible with their distant common ancestry with Aymara. In our previous WGS analysis of Andeans, we showed that windows in the *NFKB1* region were highly differentiated relative to lowland Native South Americans consistent with positive selection[10] with a peak PBSn1 value of 0.3107 ranked in the top 0.1 percentile genome-wide, which suggests the broader genomic region including rs230511 is significantly differentiated in Aymara from Native South American lowlanders (Fig. 3e). Our previous dataset included allele frequencies for three of the five sQTL intronic SNPs, with rs230511 being the most differentiated SNP in this region (Fig. 3e), suggesting that this SNP may be the target of selection. Therefore, we observed rs230511-*T* to be enriched in the Aymaras compared to general populations, Tibetans, and a different population living in the same region and used it for further analyses.

### Increased inflammatory protein and transcript levels correlate with alternative splicing of *NFKB1*

To evaluate the hypothesis that AS-*NFKB1* variants may affect the inflammatory response, we next measured canonical transcript levels of the inflammatory genes *IL6* and *IFNG* (i.e., two target genes of NF-

kB) in granulocytes, along with their plasma protein levels (i.e., IL6 and IFN-γ). Both transcripts showed significant association with the rs230511 *T* allele (i.e., the Aymara enriched allele) (Fig. 4a, d), and their plasma protein levels were also significantly associated in the same direction (Fig. 4c, f). The transcript levels of these target genes were positively correlated with AS-*NFKB1* transcript levels, but weakly and inversely correlated with canonical *NFKB1* transcript levels, though the correlation was not statistically significant (Fig. 4b, e).

We then searched for correlations between dysregulated expression of inflammatory genes and AS-*NFKB1* transcripts. Among the 60 inflammatory genes upregulated in Aymara, 33, 58, and 59 genes were positively correlated with *NFKB1-AS1*, *NFKB1-AS2*, and *NFKB1-AS3*, respectively. Ten genes of these 60 genes were inversely correlated with canonical *NFKB1,* but positively correlated with *NFKB1-AS2* and *NFKB1-AS3*. Among 16 genes downregulated in Aymara, only 1 and 2 genes were inversely correlated with *NFKB1-AS2* and *NFKB1-AS3*, respectively. Ten genes of these 16 genes were positively correlated with canonical *NFKB1* transcript levels. Among these downregulated genes, only two genes (*CORO1A* and *PYCARD*) were inversely correlated with *NFKB1-AS3* but positively correlated with canonical *NFKB1*. Only 16 genes among the 76 dysregulated inflammatory genes were correlated in opposite directions by *NFKB1-AS2*, *NFKB1-AS3*, and canonical *NFKB1* (Supplementary Data 8).

### Adaptive targets of NF-kB-regulated genes

We explored the possibility that AS-*NFKB1* may directly modulate the transcript levels of genes regulated by NF-kB. Ingenuity Pathway Analysis (IPA) analysis indicated activation of the NF-kB complex as an upstream regulator (Supplementary Fig. S4), suggesting that NF-kB target genes are likely to be upregulated in Aymara. Using RNA-seq data, we examined the expression of NF-kB target genes in Aymara granulocytes. Among 60 dysregulated NF-kB target genes, transcripts

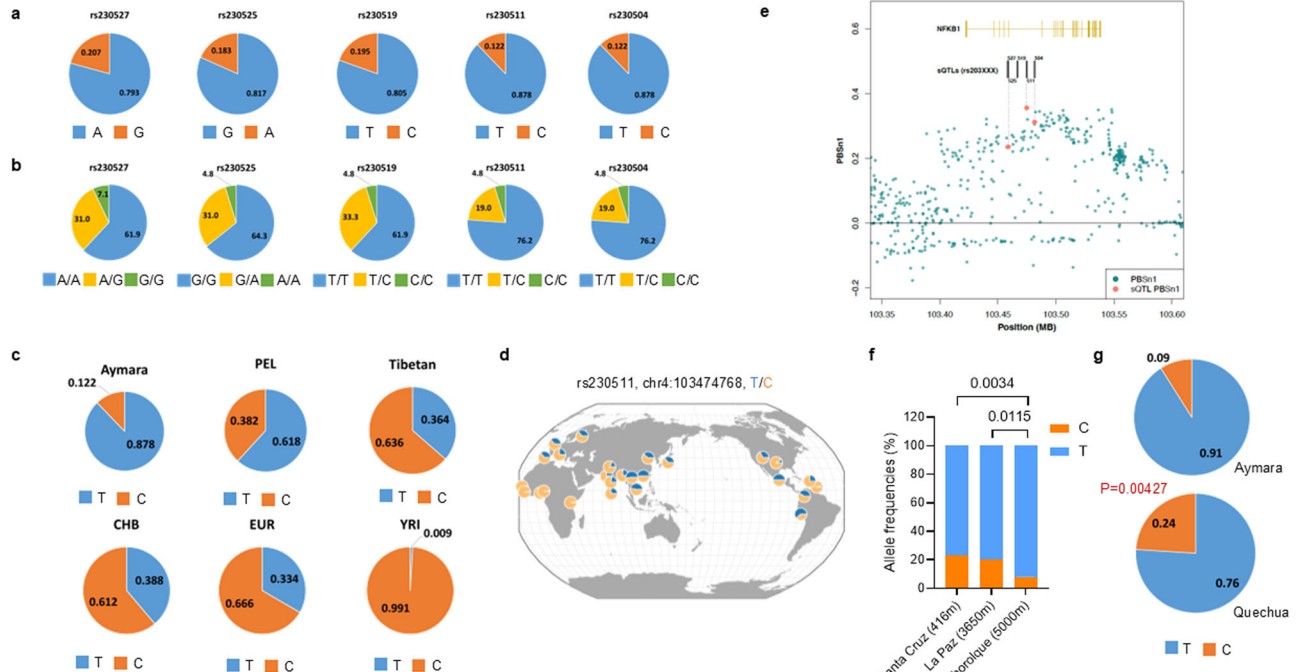

**Fig. 3 | Five Aymara enriched *NFKB1* SNPs. a** Allele frequencies in the Aymara population. **b** Genotype frequencies in the Aymara population. **c** Allele frequencies of rs230511 in Aymara, Peruvians from Lima, Peru (PEL), Tibetans, Han Chinese in Beijing, China (CHB), Europeans (EUR), and Yoruba in Ibadan, Nigeria (YRI). **d** Geography of rs230511 alleles provided from the Geography of Genetic Variants Browser[28]. **e** Selection signals of Aymara enriched *NFKB1* SNPs, calculated from F12 values that were identified as Fst between Aymara and lowland Native American

ancestry component and PBSn1 calculated by normalizing the PBS statistic for 10 Aymara-enriched SNPs[10]. **f** Allele frequency of rs230511 in Aymara living at three different altitudes ($n = 43$ from Santa Cruz at 400 m, $n = 133$ from La Paz at 3650 m, and $n = 12$ from Chorolque at 5000 m). **g** Allele frequencies of rs230511 in Aymara ($n = 12$) and Quechuas ($n = 49$) living at 5000 m. *P* values in **f** and **g** were calculated using the two-tailed Chi-squared ($\chi^2$) test.

of 51 genes were upregulated and those of 9 genes were down-regulated (Adj. $p < 0.05$ and Log$_2$ fold change >0.5).

Among the 51 upregulated genes in Aymara, 38 had expression levels that positively correlated with *NFKB1-AS1. NFKB1-AS2* and *NFKB1-AS3* were positively correlated with 53 and 52 genes, respectively. Among these 51 genes, 6 genes (*CCR7, MYC, CDC25B, CD40LG, IFNG, CLU*) were inversely correlated with canonical *NFKB1*.

Among the 9 downregulated genes in Aymara, none of these genes were correlated with AS-*NFKB1* transcripts but *P2RY2, FCGRT, CXCL6* were positively correlated with canonical (Supplementary Data 9).

### Adaptive changes of HIF target genes

Since NF-kB also regulates HIFs[29,30], our transcriptome analysis also predicted that NF-kB increases HIF-1α expression (Supplementary Fig. S4), and HIF-1 and HIF-2 signaling pathways were upregulated in Aymara (Supplementary Fig. S5), we next analyzed changes in HIF-regulated gene expression using whole transcriptome data. Among 46 dysregulated genes, 41 were upregulated and 5 downregulated in Aymara compared to Europeans (Adj. $p < 0.05$ and Log$_2$ fold change >0.5). Out of the 41 upregulated genes, 21, 40, and 40 were positively correlated with *NFKB1-AS1, NFKB1-AS2,* and *NFKB1-AS3*, respectively, while 7 genes were inversely correlated with canonical *NFKB1*. Among 4 downregulated genes in Aymara, *CORO1A,* and *NCOA4* were inversely correlated with *NFKB1-AS2* and *NFKB1-AS3* but positively correlated with canonical *NFKB1. BID* showed a positive correlation with canonical *NFKB1* but no correlation with AS-*NFKB1* transcripts (Supplementary Data 10).

Our analysis identified 14 genes (*CD44, IFNG, TNFAIP3, F3, SNAI1, CCL5, SLC7A5, TNF, MYC, AHR, CDKN1A, SMAD7, BACH2*, and *BIRC2*) regulated by both HIFs and NF-kB (Adj. $p < 0.05$ and Log$_2$ fold change >0.5) and all genes' transcripts except *BIRC2* were

upregulated in Aymara. These genes were positively correlated with *NFKB1-AS2* and *NFKB-AS3*. Except *IFNG* and *CCL5*, other genes displayed positive correlations with *NFKB1-AS1. IFNG* and *MYC* were negatively correlated with canonical *NFKB1* (Supplementary Data 11).

We further confirmed the changed expression of *CDKN1A, TNF, BIRC2* (three genes regulated by both HIF and NF-kB), *BCL2* (NF-kB target gene), *EDN1, PTGS2, SOD2* (HIF target gene) transcripts by qRT-PCR (Supplementary Fig. S6).

### Correlation of Aymara-enriched *NFKB1* SNPs with alternative spliced *NFKB1* transcripts and high Hb

We then evaluated the presence of correlations between Aymara-enriched *NFKB1* SNPs and Hb levels. Individuals with C/T and T/T genotypes at rs230511 had significantly higher Hb levels relative to C/C controls (Fig. 5a) and canonical *NFKB1* transcript levels inversely correlated with high Hb. Hb levels correlated positively with *NFKB1-AS2* and *NFKB1-AS3*: *NFKB1-AS3* was observed to have the strongest association with Hct ($r = 0.4125, p = 0.0291$); (Fig. 5d). Among the other four SNPs less enriched in Aymara, only those heterozygous at any of three SNPs (rs230527, rs230525, and rs230519) had high correlation with high Hb compared to controls (data not shown).

### Aymara-enriched *NFKB1* SNPs and alternatively spliced *NFKB1* transcripts correlate with white blood cell and platelet counts

Individuals homozygous for the Aymara enriched allele (T/T) at rs230511 had higher white blood cell (WBC) and platelet count compared to those with C/T genotype (Fig. 5b, c). Furthermore, these cell counts positively correlated with all AS-*NFKB1* transcript levels: *NFKB1-AS2* was observed to have the strongest association (Fig. 5e, f). However, canonical *NFKB1* transcript levels were inversely correlated with platelet counts but not with WBC counts (Fig. 5e, f).

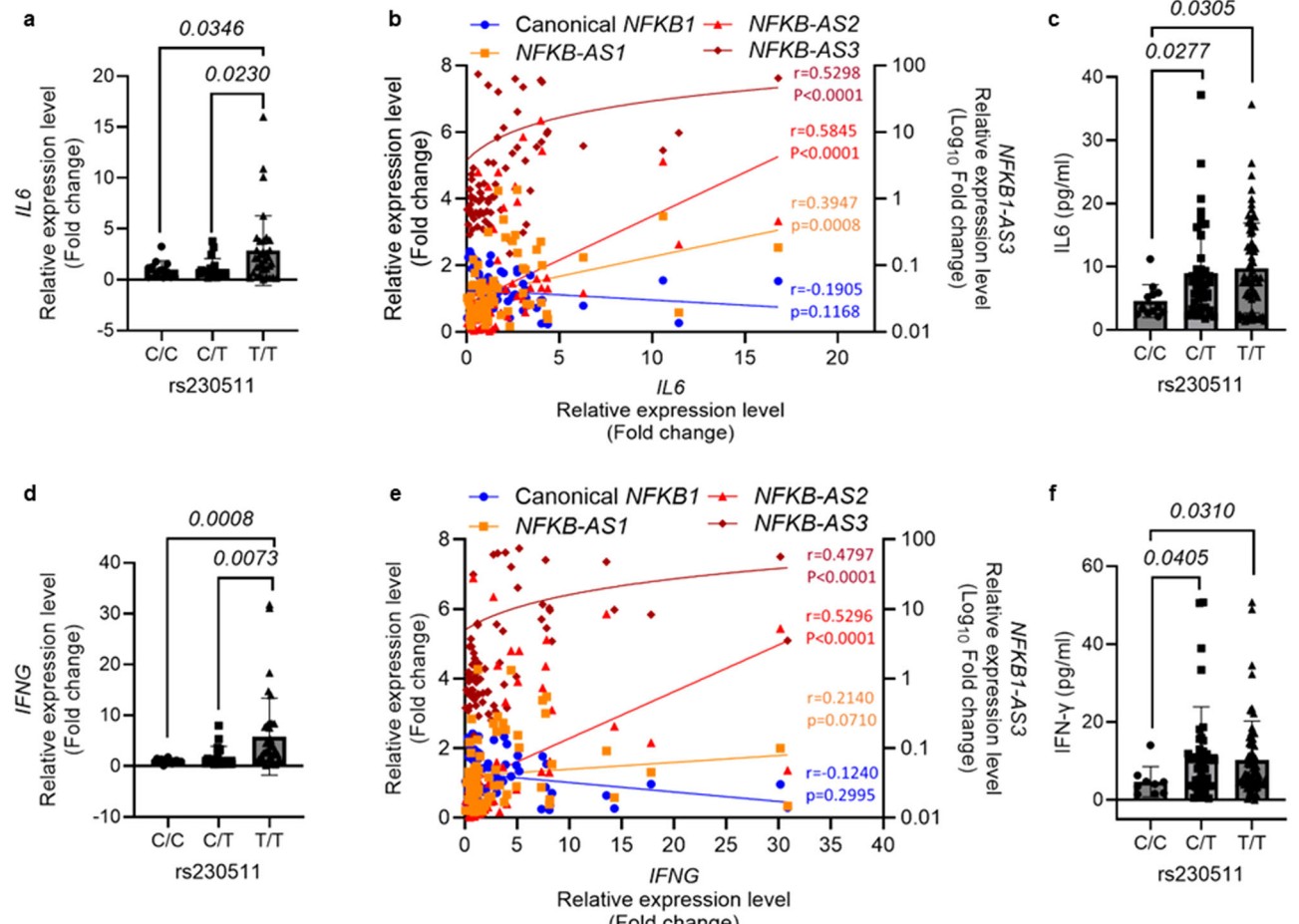

**Fig. 4 | Positive correlation of *NFKB1* SNP rs230511 genotype and AS-*NFKB1* with IL-6 and IFN-γ transcript and protein levels. a** *IL6* transcript levels in granulocytes and **b** their correlation with canonical *NFKB1*, *NFKB1-AS1* (Exon 4 skipped AS-*NFKB1*), *NFKB1-AS2* (Exon 5 skipped AS-*NFKB1*), and *NFKB1-AS3* (Exon 4 and 5 skipped *AS-NFKB1*). **c** Plasma IL6 protein levels according to rs230511 genotype. **d** *IFNG* transcript levels in granulocytes and (**e**) their correlation with canonical *NFKB1*,

*NFKB1-AS1*, *NFKB1-AS2*, and *NFKB1-AS3*. **f** Plasma IFN-γ protein levels according to rs230511 genotype. C/C ($n = 13$), C/T ($n = 19$), T/T ($n = 35$), All data are expressed as mean ± standard deviation (SD). *P* values were calculated by two-tailed Mann−Whitney test. Spearman's r and *p* values of correlation analyses were calculated by GraphPad Prism 10. The *p* value was determined using a two-tailed test, and the confidence interval was set at 95%.

## AS-*NFKB1* variants cause disrupted protein translation and processing

To functionally characterize AS-*NFKB1* variants, we overexpressed N-terminal EGFP-tagged fusion constructs of canonical NFKB1/p105 and p50 proteins and corresponding AS-*NFKB1* variants in HEK293 cells. Both variants, where stop codon is generated due to deletion of exon 4 or exon 4 and 5 (AS1 and AS3), fail to translate into proteins, as evidenced by the detection of only N-terminally expressed GFP in western blots (Fig. 6a). For the AS2 variant, we consistently observed reduced expression levels of p105 and p50, along with impaired p50 processing (Fig. 6a, red arrow indicating the expected position of processed NFKB1/p50 protein). Subsequently, we analyzed cytoplasmic and nuclear extracts from HEK293 cells transfected with NFKB1/p105 canonical (WT) and AS2 variants. While canonical endogenously processed NFKB1/p50 was detected in both fractions (Fig. 6b), NFKB1/p50 AS2 protein was only evident in the cytoplasmic lysate after prolonged exposure (Supplementary Fig. S7). Despite these observations, the reporter gene transcription assay revealed that processed NFKB1/p50 AS2 exhibited a comparable inhibitory effect on RelA-dependent promoter activation as the WT, whereas NFKB1/p50 AS1 and AS3 variants showed no impact on promoter activity (Fig. 6c). Overall, AS-*NFKB1* transcripts cause complete (AS1 and AS3) and partial (AS2) loss of canonical NFKB1 function.

## AS-*NFKB1* overexpression induces NF-κB protein but reduces it under inflammatory stress

We investigated whether AS-*NFKB1* overexpression affects NF-κB protein levels in granulocytes and its targeted inflammatory gene expression levels. We overexpressed three AS-*NFKB1* isoforms in HL60 cells, a promyelocytic leukemia cell line that serves as a model for granulocytes. Cytoplasmic and nucleus NF-κB protein levels were measured both with and without TNF treatment in conditions of AS-*NFKB1* overexpression. Cytoplasmic NF-κB protein levels did not differ significantly in response to AS-*NFKB1* overexpression or TNF treatment (Fig. 7a and Supplementary Fig. S8). However, TNF treatment promoted NF-κB translocation into the nucleus. Compared to WT-*NFKB1* overexpression, AS3 overexpression increased NF-κB protein levels. However, with TNF treatment, AS3 overexpression led to a reduction in NF-κB protein levels in the nucleus (Fig. 7b and Supplementary Fig. S8). We then measured the expression levels of inflammatory genes regulated by NF-κB. Overexpression of AS3 induced the expression of inflammatory genes such as *IL1B*, *TNF*, and *CDKN1A* (Fig. 7c and Supplementary Fig. S9). HIF-target genes, including *SLC2A1*, *VEGFA*, and *EPAS1* were also upregulated compared to cells overexpressing WT-*NFKB1*. However, under conditions of heightened inflammatory stress (induced by TNF treatment), the expression levels of *SLC2A1*, *VEGFA*, *LDHA*, and *BCL2* with AS3

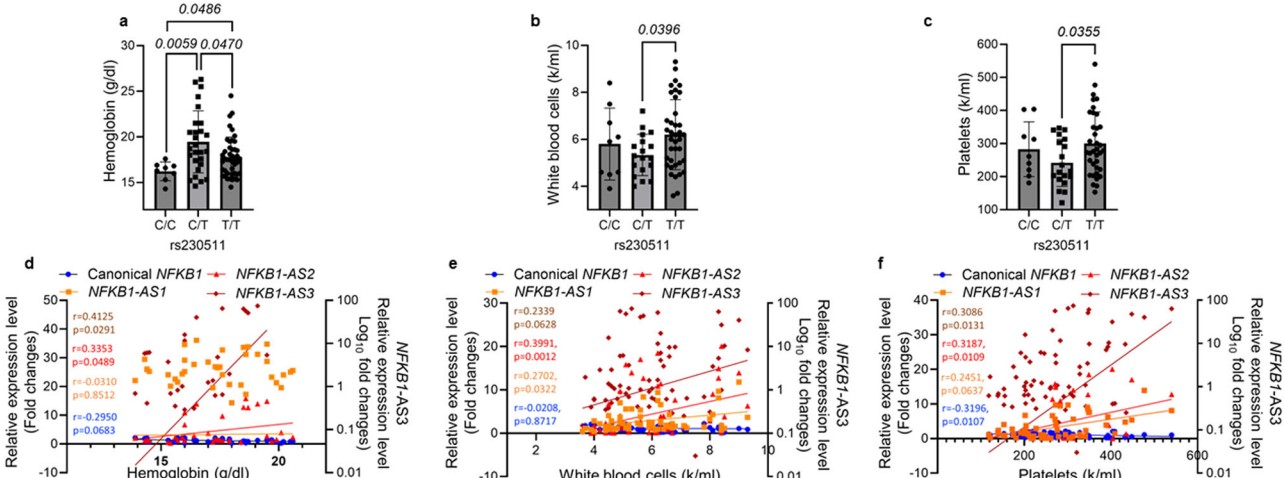

**Fig. 5 | Positive correlation of *NFKB1* SNP rs230511 genotype and alternatively spliced *NFKB1* transcripts with hemoglobin, white blood cell count, and platelet count. a** Hemoglobin (Hb) ($n = 8$ for C/C, $n = 28$ for C/T, $n = 49$ for T/T), **b** white blood cell count ($n = 9$ for C/C, $n = 18$ for C/T, $n = 39$ for T/T), and **c** platelet count ($n = 9$ for C/C, $n = 19$ for C/T, $n = 37$ for T/T) according to rs230511 genotype. Correlation of canonical *NFKB1*, *NFKB1-AS1* (Exon 4 skipped AS-*NFKB1*), *NFKB1-AS2* (Exon 5 skipped AS-*NFKB1*), and *NFKB1-AS3* (Exon 4 and 5 skipped *AS-NFKB1*) with **d** Hb, **e** white blood cell count, and **f** platelet count. All data are expressed as mean ± standard deviation (SD). *P* value was calculated by two-tailed unpaired *t*-test. Spearman's *r*-values of correlation analyses were calculated by GraphPad Prism 10. The *p*-value of correlation analyses was determined using a two-tailed test, and the confidence interval was set at 95%.

overexpression were reduced (Fig. 7d and Supplementary Fig. S9). These results correlated with decreased expression of NF-κB protein levels in the nucleus.

### *NFKB1* SNP correlated with lower inflammatory gene expression in patients with high baseline inflammation

Overexpression of AS-*NFKB1* experiment showed decreased or no changes of inflammatory and HIF-targeted gene expression under high inflammatory stress. We tested if inflammatory gene expression levels in the patients with high inflammation correlate with *NFKB1* SNP rs230511. We collected granulocytes and platelets of polycythemia vera (PV) and essential thrombocythemia (ET), which are acquired blood diseases with high baseline inflammation compared to healthy controls[31]. We genotyped 30 PV and 15 ET patients and assessed the expression of inflammatory genes (*IL1B*, *CXCL8*, *IL6*, *IL15*, and *TNF*) in granulocytes and platelets. In platelets, individuals with the C/T genotype showed lower transcript levels of these inflammatory genes compared to those with the C/C genotype, while *IL1B*, *CXCL8*, and *TNF* were significantly lower in the T/T genotype. In granulocytes, *IL1B* and *CXCL8* expressions were reduced in the T/T genotype, whereas *IL15* and *TNF* were decreased in the C/T genotype. We then analyzed HIF-target genes (*LDHA*, *SLC2A1*, and *VEGFA*), which were significantly lower in C/T genotypes in both PV and ET granulocytes and platelets (Supplementary Fig. S10).

## Discussion

In this study, we conducted integrative analysis of WGS and RNA-seq data measured from granulocytes of Andean Aymara and European samples to explore a molecular signature of Aymara adaptation to high-altitude. Since granulocytes are abundant in whole blood and release pro-inflammatory cytokines, chemokines, and other mediators, they play a key role in initiating and sustaining the inflammatory process, leading to our decision to study these cells. Although high Hb levels can be detrimental, Aymara exhibits a distinct evolutionary adaptation with high Hb, unlike high-altitude Tibetans and Ethiopians, which maintain Hb levels similar to those of lowlanders. Our previous WGS analysis found evidence for natural selection at a number of genomic regions, but the strongest signals did not include genes that would explain high Hb in Aymara as an evolutionary strategy to improve oxygen delivery through induced

erythropoiesis[10]. Therefore, we employed a strategy integrating genetic and transcriptional regulation to investigate the genetic underpinnings of Aymara phenotypes, positing that WGS analysis may not provide a complete picture of the genetic signature of this adaptation. We have also used a higher resolution than has been performed in other integrative studies[32,33].

Our results revealed transcriptional perturbations (i.e., DEGs and DSGs) and associated *cis*-genetic regulators (eQTLs and sQTLs) of immune-, inflammatory-, and hypoxic-related pathways and thereby potentially contribute to distinct erythropoiesis regulation in the Aymara. We discovered AS-*NFKB1* transcripts that skip exon 4 and/or exon 5 encoding the *NFKB1* DNA-binding domain. *NFKB1* was considered a strong candidate gene to explain Aymara adaptation since (1) it is a hub gene for inflammation and hypoxia pathways in our PPI network and (2) the AS-associated SNP sQTL, rs230511-*T* was enriched in the Aymara population relative to other populations.

*NFKB1* is well-documented to play the regulatory role in functional pathways of inflammation, hypoxia, and erythropoiesis. It encodes the protein p105, which is processed into the shorter protein p50. P50 generates heterodimeric NF-kB complexes by interacting with p65 (RELA), c-Rel, or RelB. Inhibitors of NF-kB (IkBs) bind to these heterodimers. In the presence of appropriate stimuli, IkBs are phosphorylated by IkB kinase (IKK) and then degraded by the proteasome. Uninhibited, the NF-kB heterodimer complex is translocated into the nucleus and induces expression of genes encoding proinflammatory cytokines, chemokines, adhesion molecules, and anti-apoptotic factors that are associated with inflammation[34,35]. NFKB1 functions as a potent inflammation suppressor allowing p50 to form a homodimer that acts as a transcriptional repressor[36]. *Nfkb1* knockout mice exhibit increased inflammation and tissue neutrophil infiltration associated with elevated chemokine and cytokine production[37]. Non-functional AS-*NFKB1* transcripts correlate with elevated levels of inflammatory transcripts and proteins, whereas canonical *NFKB1* transcript levels exhibit an inverse relationship with these measurements (Fig. 4b, e), providing further evidence for the inhibitory role of NFKB1 in inflammation, thus its loss-of-function transcripts have opposite function.

Additionally, NF-kB is known as an inducer of *HIF1A* (encoding HIF-1α) gene transcription; HIFs α subunits determine the levels of HIFs dimers and HIFs transcriptional activity. HIF-1 also regulates NF-kB[38]. Under hypoxia, HIF-1 induces cell proliferation via activation of

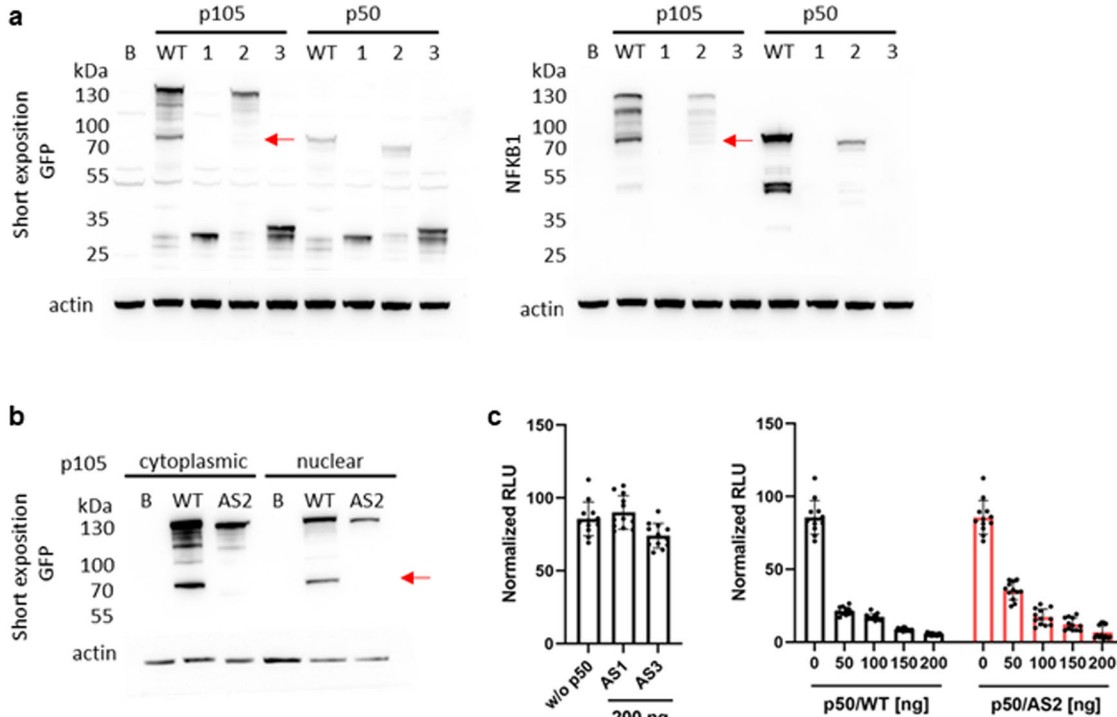

**Fig. 6 | Disrupted protein translation and processing of AS-*NFKB1* variants.** HEK293 cells were transiently transfected with EGFP-fusion constructs either encoding canonical NFKB1/p105 and p50 forms (WT) or AS-*NFKB1* transcripts (AS1, 2, 3). **a** Whole cell lysates were analyzed by western blotting using antibodies against GFP (left) and NFKB1 (right), β-actin was used as loading control. **b** Cytoplasmic and nuclear extracts of HEK293 cells transiently transfected with EGFP-fusion constructs of canonical NFKB1/p105 and NFKB1-AS2 (GFP antibody was used and β-actin as loading control). **c** The inhibitory effect on RelA-dependent promoter activation is not mediated by NFKB1/p50 AS1 and AS3 variants (left),

whereas NFKB1/p50 AS2 has comparable inhibitory effect as WT (right). HEK293 were co-transfected with synthetic reporter gene composed of an NF-κB responsive promoter driving luciferase expression. Reporter expression is switched on to maximum by co-expression of RelA (p65, 50 ng). The different concentrations of NFKB1/p50 WT and AS2 were added to inhibit the RelA-mediated reporter activation, maximum concentration (200 ng) was used for NFKB1/p50 AS1 and AS3. We performed at least three independent experiments. Abbreviations: B−background (not transfected HEK293 cells), red arrow indicates expected position of processed NFKB1 protein. All data are expressed as mean ± standard deviation (SD).

NF-kB[39]. Decrease of *HIF1A* transcripts with siRNA suppresses NF-kB activity by reducing its DNA binding activity[40]. In the presence of external stimuli such as lipopolysaccharide and IFN-γ, the NF-kB complex increases HIF-1α protein levels[41]. In addition, inhibition of NF-kB complex activity by pyrrolidine dithiocarbonate decreases *HIF1A* transcript levels in fibroblasts[42]. Overexpression of two subunits of NF-kB (p50 and p65) has been shown to enhance *HIF1A* transcription, while inhibition of NF-kB attenuates *HIF1A* transcription induced by hypoxia[43] and also increases expression of *EPAS1*, which encodes HIF-2α, principal inducer of *EPO* transcription[44]. An *EPAS1* promoter binding assay identified RELA, a NF-kB family member, as a potent inducer of *EPAS1* expression[45]. As expected, the activated NF-kB complexes induced *HIF1A* transcription (Supplementary Fig. S4), and HIF-1 signaling pathway was upregulated in Aymara (Supplementary Fig. S5).

Finally, NF-kB signaling also regulates erythropoiesis. NF-kB activity is higher in early erythroid progenitors, leading to down-regulation of genes required for differentiation. During differentiation, NF-kB activity decreases, allowing the expression of NF-E2, an ery-throid transcriptional factor[46,47]. Mice lacking *RelB*, a member of NF-kB complexes, have impaired erythropoiesis in the bone marrow, which is compensated by increased hematopoiesis in the liver and spleen, resulting in normal Hb levels[48]. Using in vitro erythroid expansion of erythroid progenitor cells from peripheral blood, we confirmed that *NFKB1* and AS-*NFKB1* transcript levels also decreased during erythroid differentiation (data not shown). However, due to the paucity of material, we could not study the changes in differentiation, proliferation, and apoptosis of erythroid cells with and without the AS-*NFKB1* transcripts.

We set out to validate the results from our integrative analysis by conducting a comprehensive functional analysis. We validated the splicing events in *NFKB1* and their genetic regulator, rs230511, as a sQTL in a region that shows high levels of genetic differentiation between the Aymara population and lowland Native Americans, consistent with recent positive selection of this genomic region (Fig. 3e). We then evaluated gene frequency of rs230511 SNP in our Tibetan and European genomes database[14,49] and found rs230511-*T* at lower allele frequencies in these two populations than in Aymaras. We also genotyped Quechuas, another high-altitude native Andean population distantly related to Aymaras, living at Chorolque (~5000 m) and found lower rs230511 allele frequency than Aymaras (Fig. 3g). Moreover, genotyping of Aymaras living at different alti-tudes revealed that this variant allele is more frequent with increas-ing altitude, consistent with a selective advantage at high altitude. Our findings indicated that AS-*NFKB1* decreased inflammatory responses under inflammatory stress although it is positively corre-lated with inflammation in the normal state. The decreased inflam-matory responses in Aymara population might help in adaptation to high altitude that could contribute to alleviation of chronic inflammation[50–52]. Taken together, these results suggest that rs230511 is the target of natural selection in this genomic region and underlies, at least in part, the genetic and physiological adaptations to the high altitude in the Aymaras.

Our functional study identified AS-*NFKB1* transcripts as positively correlating with not only transcript and protein levels of the inflam-matory genes *IL6* and *IFNG* (encoding interferon γ), but also with Hb levels, white blood cells, and platelets, demonstrating that these

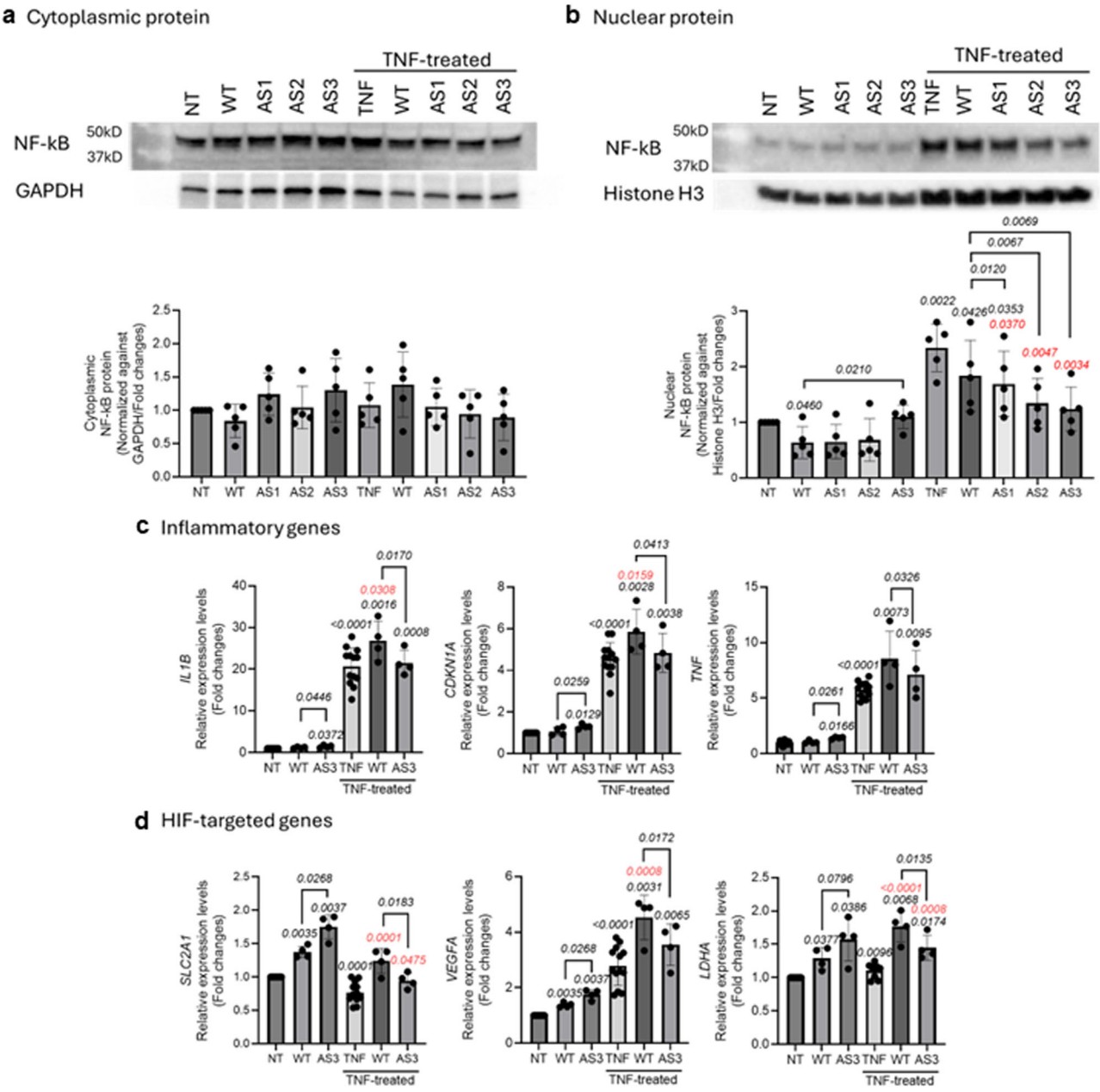

**Fig. 7 | Under inflammatory stress, *NFKB1-AS3* decreased translocation of NF-kB into nucleus results in reducing expression levels of inflammatory and HIF-targeted genes. a** Cytoplasmic NF-κB and **b** nuclear NF-κB protein levels were measured in three AS-*NFKB1* or canonical *NFKB1*-overexpressed HL60 with and without TNF treatment. Measured cytoplasmic or nuclear NF-κB protein levels were normalized against GAPDH or Histone H3, respectively. We conducted five independent Western blot experiments under identical experimental conditions. *P* values were calculated using a two-tailed paired *t*-test. The expression levels of **c** Inflammatory (*IL1B*, C*DKN1A*, *TNF*) and **d** HIF-targeted genes (*SLC2A1*, *VEGFA*,

*LDHA*) were measured in the same cells and normalized against *RPL13A* transcript levels. We conducted four independent overexpression experiments under identical experimental conditions. *P* values were calculated using a two-tailed paired *t*-test (for comparisons to NT or between WT and AS) or an unpaired *t*-test (for comparisons to TNF). Black *P* value compared to NT; Red *P* value compared to TNF; All data are expressed as mean ± standard deviation (SD). Abbreviations: NT; no-treatment, WT; canonical *NFKB1*, AS1; *NFKB1*-AS1, AS2; *NFKB1*-AS2, AS3; *NFKB1*-AS3, TNF; TNF treatment.

evolutionary selected genetic variants account for Aymara high Hb. Notably, although inflammation suppresses erythropoiesis through increasing hepcidin, the master regulator of iron metabolism[53], we found a strong correlation of Hb level in Aymara with *NFKB1* Aymaras' evolutionary selected SNPs and AS-*NFKB1* transcripts. However, AS-*NFKB1* also positively correlated with HIF-regulated genes. This indicates that the upregulation of HIFs augments erythropoiesis and overcomes the inhibitory effects of inflammation on erythropoiesis. Other evidence suggests that high-altitude-induced hypoxia may also prolong the lifespan of red blood cells[54,55], which might be an

additional causative mechanism of erythrocytosis in Aymara[56] independent of the NF-kB pathway.

We also tested whether AS-*NFKB1* was translated into protein and altered NF-κB protein levels (Figs. 6a–c, and 7). Only AS2 generated p50 (an NF-κB protein) but had a defect in nuclear translocation. Overexpression of AS-*NFKB1* did not affect cytoplasmic p50 protein levels; however, nuclear NF-κB protein levels were altered with AS-*NFKB1* overexpression or inflammatory stress induced by TNF treatment. These data suggest that the nuclear translocation of NF-κB protein may be influenced by AS-*NFKB1*. Inflammatory gene

expression under these conditions correlated with these protein levels. We also confirmed these changes in disease models, including PV and ET, which are characterized by increased baseline inflammation. In these patients, the *NFKB1* rs230511 (C/T or T/T genotype) was associated with lower inflammatory gene expression. These findings suggest that AS-*NFKB1* may be beneficial under inflammatory stress by preventing excessive expression of inflammatory genes.

Higher WBC counts also correlate with increased inflammation[57], as well as platelet counts, that positively correlate with inflammatory proteins plasma levels including C-reactive protein (CRP) and cytokines[58]. We also found a positive correlation of AS-*NFKB1* transcript levels with WBC and platelet counts, which suggests a possible role for these transcripts in Aymara by upregulating the inflammatory response under hypoxic conditions.

Although we found only a small number of genes that were simultaneously DEGs and DSGs, enriched pathways were largely shared between the two differential gene sets for immune-related functions. This suggests that a complex interconnection of gene expression and splicing may contribute to the genomic and transcriptomic regulation that results in the distinct Aymara molecular signatures resulting in their beneficial physiological adaptation to extreme environmental hypoxia. We present one such mechanism here: *NFKB1* alternative splicing. Differential splicing of its exons 4 and 5 could affect the functional levels of NFKB1, HIF-1, and immune-related genes. The altered splicing of this upstream regulator changes the expression of its downstream target genes that contribute to Aymara adaptations. This observation could explain why there is a significant overlap in gene sets between DEGs and DSGs, but less similarity at the level of individual genes. We then found the genetic signature of *cis*-SNPs that act as sQTLs and regulate these splicing events. Notably, these SNPs are considered both *cis*-sQTLs and *trans*-eQTLs since the associated splicing status could also affect downstream target gene expression.

Firstly, there are some limitations of our study that could lead to potential overinterpretation of these findings. In our pilot studies, we started with a relatively small sample set that may result in a possibility of missing some genes that also contribute to Aymara adaptation. Therefore, further study with more samples is required to obtain a more comprehensive perspective and replicate our computational analysis results. Additionally, gene regulation can differ across various cell types, so while focusing on granulocytes provided specific insights, this approach may not capture the full complexity of gene regulation across other blood cell types. The challenges of separating cells from whole blood in Bolivia further restricted our ability to obtain pure populations of platelets and other blood cells, limiting the scope of our analysis to granulocytes. Secondly, AS-*NFKB1* variants showed loss of function as repressors of NF-κB pathway, nevertheless, NFKB1/p105 have also non-canonical function[59], which remains to be determined. Also, in our study, the increased AS-*NFKB1* transcript level was associated with high NF-kB transcriptional activity and the possible role of the non-coding transcript variants of protein-coding genes needs to be further explored. While we demonstrated that AS-*NFKB1* reduced NF-kB translocation to the nucleus and downregulated inflammatory and HIF-targeted gene expression under inflammatory stress, the underlying mechanisms remain to be fully elucidated. Thirdly, although our findings indicate relationships between AS-*NFKB1* and inflammation, as well as HIF-targeted genes, suggesting that high Hb levels in Aymara individuals may be, at least in part, mediated by upregulation of HIFs via the loss of function of AS-*NFKB1* transcripts. However, the complete mechanism underlying AS-*NFKB1*'s regulation of erythropoiesis in this population requires further characterization in future studies.

It will be necessary for follow-up experimental studies to confirm our other computational results and elucidate other Aymara-specific molecular mechanisms, as the scope of the deep analysis in this work was largely limited to *NFKB1*. However, our data provides the first candidate for Aymara increased Hb in high altitude and their possible protective mechanisms against excessive inflammation.

## Methods
### Study subjects
The initial ten Aymara and four European samples were collected at La Paz-Bolivia (3650 m) its suburb El Alto is at 4150 m. This was followed by validation samples of 55 Aymara and 18 Europeans living in La Paz or its higher-located suburb, El Alto. Subsequent samples were collected from Bolivian Aymara living at different altitudes: $n = 43$ from Santa Cruz (400 m), $n = 133$ from La Paz (3650 m), and $n = 12$ from Chorolque (5000 m). Additionally, $n = 49$ Quechua individuals from Chorolque were recruited by Dr. Ricardo Amaru. The study was approved by the Institutional Review Board of San Andres University Medical School, La Paz, Bolivia, and all participants provided informed consent. Peripheral blood was collected using ethylene diamine tetra acetic acid (EDTA) tubes and complete blood count was performed by automated hematological counter (Horiba ABX Micros ES 60, France). Since hemoglobin levels differ between males and females, we included only male participants in this manuscript to reduce variation related to menstrual cycle-caused iron deficiency and pregnancies in young otherwise healthy women.

### WGS and RNA-seq data processing
RNA-seq data was generated using the same methods as we described in Gangaraju, et al.[31], and detailed methods were as follows. mRNA sample preparation utilized the standard Illumina protocols. The Illumina TruSeq stranded RNA sample preparation kit was used for the total RNA sample libraries prepared from granulocytes. The mRNA libraries were constructed by removing ribosomal RNA from the total RNA samples with Ribo-Zero Gold oligos including beads that are complementary to cytoplasmic rRNA. The quantity and quality of RNA samples were measured on an Agilent Technologies 2200 TapeStation using the D1000 ScreenTape assay. For each of the 14 independent biological samples, RNA-seq data were generated from a HiSeq using 125 Cycle Paired-End Sequencing v4. The raw sequencing file (fastq format) contained about 30–35 million reads of 125 nucleotides in length per sample.

For WGS data, we obtained WGS data (NCBI BioProject accession number PRJNA393593) from the same samples used for RNA-seq, which were generated and analyzed in our previous study[10]. We conducted quality control (QC) for all paired-end WGS data using FastQC[60] and trimmomatic 0.32[61] with the following options: TruSeq3-PE-2.fa:2:20:10, LEADING:3 TRAILING:3 SLIDINGWINDOW:4:15, and MIN-LEN:75. Contaminant sequences (i.e., adaptor or linker sequences) were removed during QC. Clean reads were then mapped to the human genome reference sequence (GRCh37.75 based on hg19) using HISAT2. Next, we added read-group identifier (ID) information and marked duplicate reads in the mapped bam files using AddOrReplaceReadGroups and MarkDuplicates of the Picard v2.1.1 package. Finally, the reads in regions close to putative indels were locally re-aligned using IndelRealigner from the Genome Analysis Toolkit (GATK)[62] based on known indels reported in dbSNP based on hg19 reference, obtained from 1000 Genomes Project (TGP)[63]. Base quality scores were then recalibrated using BaseRecalibrator, also in GATK. From the pre-processed bam files, we then generated one single pileup result including all bam information (i.e., Aymara and European samples) using the mpileup function in SAMtools. Finally, we transformed the mapped information saved in the pileup result file into the vcf file format, including genotypes for each individual (i.e., SNPs), using VarScan v2.3.9.

For RNA-seq data, we performed quality control on each read from our raw data (fastq file) using the FastQC tool[60] the same as in the WGS analysis. We utilized RSEM to estimate gene expression based on gene annotations in GTF format (GRCh37.75) as transcripts per million (TPM) values using the STAR v2.5 aligner. We used mapped bam files (i.e., output from STAR) as input for rMATS[64], which is a tool for investigating alternative splicing of exons based on junction reads; this yielded differentially spliced exons between Aymara and Europeans. We identified all possible AS events: exon skipping (ES), alternative 5′ splicing site (A5SS), alternative 3′ splice site (A3SS), intron retention (IR), and mutually exclusive exon splicing (MXE). rMATS calculates the expression of each AS exon, represented as the percent spliced in (PSI, range zero to one) based on the reads mapped onto a boundary between consecutive exons (called junction reads). The PSI indicates the fraction of alternatively spliced exons. For example, with an exon-skipping event, a PSI of zero means that all mRNAs from the gene skipped the AS exon, while the value of one indicates that all mRNAs have the AS exon.

## Differentially expressed and alternatively spliced (AS) genes in Aymara

After estimating the gene expression and PSI level of each AS exon, we compared them between Aymara and European samples. DESeq2 was employed to compare gene expressions, and the rMATs comparison function was used to compare PSI levels. In both comparisons, $P$ values were corrected according to the false discovery rate (FDR). Differentially expressed genes (DEGs) and spliced exons (DSEs) were considered significant at FDR < 0.05, in conjunction with an absolute fold change >1.5 for gene expression levels and a difference of PSI > 0.1 (10% difference in PSI between the two populations) for AS PSI levels.

## Gene set enrichment and network analysis

For functional interpretation of our identified DEGs and DSEs, we performed gene set enrichment analysis for canonical pathways using ConsensusPathDB (CPDB; http://cpdb.molgen.mpg.de/)[65]. The cutoff value for significance was FDR < 0.05. We also constructed a protein-protein interaction network from the identified DEGs and DSEs using the StringDB web tool[66] to further gain insight into their molecular functions at the systems level. We selected only the first interaction relationship between nodes (i.e., genes) with two high confidence levels at >0.7 and excluded disconnected nodes from the network.

## Cis regulators for DEGs and DSEs

For each DEG and DSE, we performed association tests with all possible genotypes of a given SNP using a linear regression model; SNPs identified as associated are referred to as expression quantitative trait loci (eQTLs) and splicing quantitative trait loci (sQTLs), respectively. To identify eQTLs, gene expression as TPM was tested for association with SNPs located in promoter regions (upstream 1500 and downstream 200 from each transcript start site), and to identify sQTLs, spliced exon levels as PSI were tested for association with SNPs within AS exons and their flanking introns. To calculate the empirical $p$ value for naïve eQTLs and sQTLs, we generated 1000 random sets of shuffled populations in which data were randomly assigned to SNPs. For each set, we performed the same analysis of eQTLs and sQTLs (i.e., linear regression) and applied the same cutoff criterion of $p < 0.05$, but with randomized genotypes for each population and PSI. Finally, we estimated 1000 $p$ values for the statistically significant eQTLs and sQTLs.

## Determination of Aymara adapted sQTLs

To evaluate adapted sQTLs in the Aymara, we compared the allele frequencies of eQTL and sQTL genotypes between our Aymara samples and non-Finnish European ancestry samples from the Genome Aggregation Database (gnomAD; v3.1.2)[67]. This database allowed us to secure reliable differences with a large number of European samples ($n = 76,156$). Differences (that is, eQTLs and sQTLs that are genetic regulators for DEGs and DSEs in the Aymara) were identified using the Chi-square test. Aymara-adapted eQTLs and sQTLs were defined with a cutoff value of FDR < 0.05 and |different minor allele frequency > 0.3.

## Calculation of genetic differentiation using *PBSn1*

Population genetic differentiation statistics (FST and PBSn1) were extracted from a previously published dataset[10]. Briefly, low-coverage whole genome sequencing data were collected using the Illumina HiSeq 2500 platform from a sample of 42 people living at high altitude with Aymara or Andean ancestry (NCBI BioProject accession number PRJNA393593). Genotype likelihoods were calculated for each individual and then merged with genotype likelihood data from a subset of the Human 1000 Genomes panel. Admixture-corrected allele frequencies were used to calculate PBSn1 at SNPs and in windows across the genome with the Andean population component as the target population. For the analysis in this work, we identified the window centered on the target sQTL SNP and present both the window and SNP-based statistics extracted from the broader, genome-wide dataset.

## *NFKB1* SNP genotyping

Genomic DNA (gDNA) was isolated from granulocytes using the Gentra Puregene Blood Kit (Qiagen). Five *NFKB1* SNPs were genotyped using TaqMan SNP genotyping assays (Thermofisher) (Assay IDs: C___804242_30 for rs230525, C__3066468_10 for rs230511, C___804227_10 for rs230504, C___804244_20 for rs230527, and C___804237_20 for rs230519).

## Measurement of AS-*NFKB1*, *NFKB1*, and inflammatory gene transcript levels

RNA was isolated from granulocytes using Tri-reagents according to the manufacturer's protocol (Molecular Research Center). cDNA was synthesized from RNA using the SuperScript™ VILO™ cDNA Synthesis Kit (Invitrogen). Primers were designed to span exon-exon boundaries: For *NFKB1*-AS1, F-5′ ACTGCCAACAGGAGAGGATTTC 3′and R-5′ TAGTT GCAGATCTTTGACCTGA 3′; For *NFKB1*-AS2, F-5′ GCACTGCCAACAGG CAGATGGC 3′ and R-5′ TAGTTGCAGATCCTGTTTAGGTT 3′; For *NFKB1*-AS3, F-5′ GGAAGGCCTGAACAAGATGTTT 3′ and R-5′ AGTTG-CAGATCCTGTTGGCAGT 3′; For canonical *NFKB1*, F-5′ GCACTGCCAA-CAGCAGATGGC 3′ and R-5′ TAGTTGCAGATCTTTGACCTGA 3′. AS-*NFKB1* and canonical *NFKB1* transcript levels were measured by qRT-PCR using TaqMan SYBR Expression Assays (Applied Biosystems). Transcript levels were measured by TaqMan expression assay (Thermofisher) with the following assay IDs: *IFNG* (Hs00989291_m1), *IL6* (Hs00174131_m1), *CDKN1A* (Hs00355782_m1), *EDN1* (Hs00174961_m1), *BCL2* (Hs04986394_s1), *TNF* (Hs00174128_m1), *PTGS2* (Hs00153133_m1), *SOD2* (Hs00167309_m1), and *BIRC2* (Hs01112284_m1). Expression levels were normalized against *HPRT1* (Hs02800695_m1) expression, calculated using $2^{-\Delta\Delta Ct}$ method, and expressed as fold change.

## Measurement of inflammatory protein levels in plasma

IL-6 and interferon-γ protein levels were measured in frozen plasma using the ProQuantum Immunoassay Kit (Thermofisher) according to the manufacturer's instructions.

## AS-*NFKB1* cloning, western blot, and reporter assay

The AS-*NFKB1* variants were constructed using cDNA encoding N-terminally EGFP-tagged canonical NFKB1/p105 and NFKB1/p50 (kind gift of M. Fliegauf and B. Grimbacher, University of Freiburg). The DNA fragments spanning deletion of exon 4, and exon 5 and both were commercially synthesized by Integrated DNA Technologies, Inc. (IDT),

Coralville, Iowa. With unique restriction sites (BglII and Bst1107I), the wild-type sequence was replaced. Final constructs were subjected to whole plasmid sequencing (Eurofins Genomics). HEK293 (human embryonic kidney, Cat. ATCC-CRL-3249) cells were grown in Dulbecco's Modified Eagle Medium (DMEM) supplemented with 10% fetal bovine serum and 1% penicillin/streptomycin. The HEK293 cells were transiently transfected with NFKB1/p105 (or p50) WT, AS1, AS2, and AS3 plasmids using Lipofectamine 2000 (Thermo Fisher Scientific). Twenty-four hours post-transfection, 1. crude cell lysates were prepared and separated on gradient 4–20% SurePage gels (GenScript) for western blots using rabbit anti-GFP (#ab290, Abcam, dilution 1:1000), rabbit anti-NFKB1 (#13586, Cell Signaling, dilution 1:500) and rabbit anti-β-actin (#ab8227, Abcam, dilution 1:10000) antibodies; 2. cytoplasmic and nuclear fractions were prepared using NE-PER™ Nuclear and Cytoplasmic Extraction Reagents (Thermo Fisher Scientific) and separated on gradient 4-20% SurePage gels (GenScript) with antibodies specified above. For luciferase-based reporter assay, HEK293 cells were transiently transfected with NFKB responsive reporter, RelA (p65), Renilla, and NFKB1 variants, at a 1:2:1:8 ratio, using Lipofectamine 2000 (Thermo Fisher Scientific). Twenty-four hours post-transfection, the luciferase production was quantified in cell lysates with Dual-Luciferase® Reporter Assay kit (Promega) on a Glomax 20/20 luminometer (Promega). Firefly:Renilla luciferase ratios were calculated for each condition in order to normalize the NFKB1 inhibitory effect for different constructs.

### AS-*NFKB1* overexpression and western blot experiment

HL60 cells (Cat. CCL-240), a promyelocytic leukemia cell line, were purchased from ATCC and cultured in RPMI 1640 medium (Thermo Fisher Scientific) supplemented with 20% FBS at 37 °C in a humidified atmosphere with 5% $CO_2$. AS-*NFKB1* constructs were used for overexpression experiments. A total of 200 ng of each construct, including canonical *NFKB1* p50, *NFKB1*-AS1, *NFKB1*-AS2, and *NFKB1*-AS3, were transfected into $0.5 \times 10^6$ HL60 cells using Lipofectamine 2000 (Thermo Fisher Scientific). After 24 h of incubation, cytoplasmic and nuclear proteins were extracted using NE-PER™ Nuclear and Cytoplasmic Extraction Reagents (Thermo Fisher Scientific) in the presence of complete, EDTA-free Protease Inhibitor (Roche). Proteins were separated on Any kD™ Mini-PROTEAN® TGX™ Precast Protein Gels (Bio-Rad) and transferred to Immun-Blot® PVDF membranes (Bio-Rad). NF-kB, GAPDH, and Histone H3 were detected using antibodies specific to NF-kB (Cat. NBP2-22178, Novus, dilution 1:1000), GAPDH (Cat. MA5-15738-HRP, Thermo Fisher Scientific, dilution 1:10,000), and Histone H3 (Cat. ab1791, Abcam, 1:10,000 dilution), respectively. Protein bands were visualized using Immobilon® ECL UltraPlus Western HRP Substrate (Sigma) and quantified using ImageJ software. Total RNA was isolated 24 h post-transfection and reverse-transcribed into cDNA following the same method described above. Transcript levels of inflammatory and HIF-targeted genes were quantified using TaqMan expression assays (Thermo Fisher Scientific) with the following assay IDs: *IL1B* (Hs01555410_m1), *CDKN1A* (Hs00355782_m1), *TNF* (Hs00174128_m1), *SLC2A1* (Hs00892681_m1), *VEGFA* (Hs00900055_m1), *LDHA* (Hs01378790_g1), *EPAS1* (Hs01026149_m1), and *BCL2* (Hs04986394_s1). Expression levels were normalized to *RPL13A* (Hs04194366_g1) using the $2^{-\Delta\Delta Ct}$ method and expressed as fold change.

### Correlation of *NFKB1* SNP rs230511 with inflammatory and HIF-targeted genes in PV and ET

Patients were recruited from clinics at the Huntsman Cancer Center and the Veterans Affairs Hospital in Salt Lake City, Utah. The study was approved by the University of Utah's Institutional Review Board, and all participants provided informed consent. Granulocytes and platelets were isolated from whole blood using Ficoll-Paque density gradient centrifugation[31]. Total RNA was extracted and reverse-transcribed into cDNA following the same method as described

above. Transcript levels of inflammatory and HIF-targeted genes were quantified using TaqMan expression assays (Thermo Fisher Scientific) with the following assay IDs: *IL1B* (Hs01555410_m1), *CXCL8* (Hs00174103_m1), *IL6* (Hs00174131_m1), *IL15* (Hs01003716_m1), *TNF* (Hs00174128_m1), *SLC2A1* (Hs00892681_m1), *VEGFA* (Hs00900055_m1), and *LDHA* (Hs01378790_g1). Expression levels were normalized to *RPL13A* (Hs04194366_g1) using the $2^{-\Delta\Delta Ct}$ method and expressed as fold change. The *NFKB1* SNP rs230511 genotype was determined using genomic DNA extracted from granulocytes, following the same method as described above.

### Reporting summary

Further information on research design is available in the Nature Portfolio Reporting Summary linked to this article.

## Data availability

The transcriptome data of granulocytes from Aymara and European individuals generated in this study have been deposited in the NCBI database under accession code PRJNA1204927. The WGS data of 42 people living at high altitude with Aymara or Andean ancestry is available in NCBI database under accession code PRJNA393593. The data that support the findings of this study are available from the corresponding author, Josef T. Prchal (josef.prchal@hsc.utah.edu), upon request.

## Code availability

Analysis codes used in this study are available on GitHub (https://github.com/hangost/AymaraSplicing).

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

## Acknowledgements

This research was supported by the National Institutes of Health under Ruth L. Kirschstein National Research Service Award 2T32HL007576-31 from the National Heart, Lung, and Blood Institute (JS), Dorothy Brown Innovation in Science (JS), and a VA merit grant (JTP). This study was also partially supported by development funding for YL from the College of Veterinary Medicine, Seoul National University. The support and resources from the Center for High-Performance Computing (CHPC) at the University of Utah are gratefully acknowledged. The support from the Cell Biology Unit, School of Medicine at University of San Andres, as well as from Julieta Luna, Silvia Mancilla, Daniela Paton, Juan Carlos Valencia, Felipe Mamani, and Emerson Cayo, is also gratefully acknowledged. This research was also supported by Next Generation EU project LX22NPO5102 and Czech Science Foundation projects 24-11730S (LL). The authors would like to thank Vladimir Korinek for help with cloning strategy and Manfred Fliegauf and Bodo Grimbacher for providing NFKB1 and RelA constructs.

## Author contributions

RA, JS, and JTP conceived the study. JS and JTP drafted the first version of the manuscript. JS conducted genotyping and measured AS-*NFKB1*, inflammatory gene, and protein levels in Aymara samples. JS designed and carried out AS-*NFKB1* overexpression experiments in vitro and assessed inflammatory gene expression levels in PV and ET samples. RA recruited the study subjects. SH and YL contributed to the conception and design of the bioinformatics analysis and discovered the alternative splicing events in *NFKB1*. SH and DK were involved in the interpretation of computational results that YL supervised. LL prepared the NFKB1 constructs, conceived the in vitro study, and edited the manuscript. TQ processed blood samples. JC analyzed selection signal of *NFKB1* SNPs. SJK performed gene expression assay for AS-*NFKB1* overexpression experiments in vitro. All authors wrote and reviewed the manuscript. YL and JTP supervised the generation and interpretation of all data.

## Competing interests

The authors declare no competing interests.
