## [Transparent Peer Review file · Nature Communications]

Alternatively Spliced NFKB1 Transcripts Enriched in Andean Aymara Modulate Inflammation, HIF and Hemoglobin

Corresponding Author: Dr Josef Prchal

Version 0:

Reviewer comments:

Reviewer #1

(Remarks to the Author)

In this paper, there is a number of noteworthy results. The DEGs from granulocytes, the e/sQTLs, the integrative analysis that the authors have performed in two different populations, namely the Aymaras in the Andes and those from European ancestry living at low altitude in Peru. The work is significant in the field as there are many unanswered questions which, if solved, can help us understand how humans adapt or do not to environmental stresses such as hypoxia. While the analysis is robust, there are certain issues that should be raised, some more problematic than others.

1. Do the authors believe that irrespective of the volume of blood (inclusive of course of the non-RBC volume), ie, whether it is large or smaller that the V/Q in the lungs is NOT affected by the total volume of the blood? If this is the case, and i do not believe it is, that one would need to determine the mechanisms that are not only related to the GWAS related to the amount of RBCs but also the plasma (non-RBC) volume.

2. Is the gene ANP32D related to HIF?

3. Are the granulocytes RNA representative of the other inflammatory cells in PBMCs?

4. While the exon skipping is a novel finding, there is in some figures such as 3, 4, 5, enormous variability and, while statistically significant, i am not sure how significant biologically?

5. One of my major issues is that most findings in the paper are correlative to NFKB and HIF. While both are important in the study in what the authors like to prove, there is no proof that is presented by the authors in terms of the importance of NFKB in terms of the adaptation and its importance in regulating in this population the RBC volume which goes to the heart of adaptation.

Reviewer #2

(Remarks to the Author)

The manuscript "Multi-Omics analysis identifies novel alternative spliced NFKB1 transcripts enriched in Andean Aymara that increase both inflammation and hemoglobin" by Song et al reports that transcript variants of key hypoxia-induced transcription factors may be responsible for population specific high altitude adaptive signatures. The authors further report that these transcript variants may be responsible for observed higher hemoglobin as well as inflammation in Andean Aymara highlanders. This concept is of particular interest to high altitude biology highlighting regulatory roles of alternately spliced transcript variants and observed adaptive features of native highlanders.

I have gone through this manuscript with keen interest and my observations are mentioned below:

1) The authors have reported transcripts for NFKB1 in Aymara highlanders. However, the (1) protein level of NFKB1 as well as (2) functionality of NFKB1 protein in Aymara highlanders remains elusive in the manuscript. Does the exon skipped transcript variants of NFKB1 results in a functional protein?

2) The manuscript does not provide any evidence that the exon skipped transcript variants of NFKB1 bind to the downstream targets more efficiently than the canonical NFKB1.

3) Figure 5 (A, B and C panels) indicate a wide variation of hemoglobin WBC and platelet levels among the studied highlanders. The statistical correlations and significance values have been derived with a handful of samples and the authors believe that this will hold true for a larger cohort of samples. With the variations reported in the manuscript (Hb: 12 – 26 g/dL, WBC: 3 – 9 k/ml and platelets 125 -550 k/ml), it is expected that the statistical correlations will become weak and non-significant in a larger cohort of samples. The authors need to provide experimental evidence for this.

4) In continuation, the authors need to elucidate the variability of Hb, WBC and platelet levels in the study group reported in Figure 5. It is apparent that the study group comprises of both CMS patients (Hb \geq 19 g/dL for women and Hb \geq 21 g/dL for men) and healthy volunteers. A suitable explanation in this aspect is required.

5) Lines 397 - 401: The authors have made a statement that Aymara highlanders residing at higher elevations have higher frequency of rs230511-T allele and this genetic adaptation confers better physiological adaptation. According to this statement, residence at higher elevations is directly proportional to higher Hb levels as well as inflammation in Andean Aymara. The authors need to clarify this statement.

6) Does inflammation is beneficial for high altitude adaptation in Andean Aymara? The authors need to elucidate how increased IL-6 and IFN- γ levels promote or help in promoting high altitude adaptation.

Reviewer #3

(Remarks to the Author)
Song et al. review

Song et al. integrate RNA-seq and whole-genome sequencing data to unravel the molecular signatures of high-altitude adaptation in the Aymara. The authors detect many differentially expressed and differentially spliced genes in granulocytes of Aymara and Europeans. By integrating e/s-QTL mapping and protein-protein interaction networks, they identified NFKB1 as a candidate hub gene, which also shows novel exon skipping events. Furthermore, Song et al. suggest that the alternative splicing of NFKB1 plays a role in regulatory changes in inflammatory protein levels, altered regulation of NFKB1 target genes, including HIFs, and increased Hb levels.

Overall, this is a very interesting study that presents multiple lines of evidence that implicate the alternative splicing of NFKB1 as a mechanism involved in high-altitude adaptation of Aymara. While I was initially a bit concerned about the relatively small sample size for the initial RNA-seq analyses, the integration of different datasets and validation of the results in a larger number of samples has ultimately convinced me that NFKB1 might play a role in high-altitude adaptation.

However, the methods section is not very clear in all parts. There seems to be a lot of information missing in the methods section regarding measurements/analyses in the extended dataset. For example, where do the haemoglobin counts come from in Fig.5, and have they been measured in the smaller or the larger dataset, and how have they been measured? In general, it is not always clear which samples were used for which analyses, making it impossible to fully judge these analyses. Thus, the methods section should be extensively reworked and expanded.

There are many sections throughout the ms that are not well written and don't make sense, making it more difficult than necessary to follow these sections. Please carefully edit the ms.

Minor comments:

- L. 107-108: This sentence doesn't make any sense ("are not found at these genes are not evolutionary selected...").
- L. 123-124: This sentence also doesn't make any sense ("might also identify previously undescribed ...")
- L.158: Incorrect punctuation/sentence structure: " in (PSI), exon usage..."
- The use of the term 'Dysregulation' (e.g. L276) suggests that something is not working correctly and has negative effects. However, that doesn't seem to be the case, as these are adaptations, and regulation has simply changed. Therefore, I would refrain from using the term 'dysregulation' in this context.
- L. 312: No information on how Hb levels were measured or what the sample size was for this analysis.
- L. 321: No information on how WBC and platelet counts were obtained and from which samples.
- L. 437-448: How does the choice of cell type potentially influence the results? It would be good to discuss this here as well.
- L. 503: I am not sure how differential gene expression was determined. In the methods section it says that a Wilcoxon-test was used but in FigS1 it says that DESeq2 was used for identifying DEGs. This should be clarified. Furthermore, I am not sure about using Wilcoxon test is the best approach for such a limited sample size (e.g. see Li et al. 2022 Genome Biology). Which test was chosen and why?

Version 1:

Reviewer comments:

Reviewer #1

(Remarks to the Author)

Reviewer #2

(Remarks to the Author)

The authors have revised the manuscript as per the suggestions/concerns raised by all the reviewers. In addition, the authors have performed additional experiments to validate the observations and queries. Overall, the revised manuscript reports the role of alternate spliced transcript variants of NFKB1 in modulating inflammation and Hb levels in Andean Aymara. These results adds additional genetic information for human adaptations to high altitude.

Reviewer #3

(Remarks to the Author)

The authors have addressed all my previous comments.

The only remaining concern I have is regarding the added data availability statement. I think the data should be made publicly available without any restrictions (e.g. uploading the RNAseq data to NCBI) unless there is a very good reason not to, as stated in the Nature Communications guidelines. Furthermore, the current statement ('upon reasonable request') is very vague and subjective. I would urge the authors to consider uploading the data for full transparency and repeatability.

Here we address the raised comments point-by-point:

Reviewer #1 (Remarks to the Author):

In this paper, there is a number of noteworthy results. The DEGs from granulocytes, the e/sQTLs, the integrative analysis that the authors have performed in two different populations, namely the Aymaras in the Andes and those from European ancestry living at low altitude in Peru. The work is significant in the field as there are many unanswered questions which, if solved, can help us understand how humans adapt or do not to environmental stresses such as hypoxia. While the analysis is robust, there are certain issues that should be raised, some more problematic than others.

1.1. Do the authors believe that irrespective of the volume of blood (inclusive of course of the non-RBC volume), ie, whether it is large or smaller that the V/Q in the lungs is NOT affected by the total volume of the blood? If this is the case, and i do not believe it is, that one would need to determine the mechanisms that are not only related to the GWAS related to the amount of RBCs but also the plasma (non-RBC) volume.

Response: Thank you for your insightful comments. We fully agree that the hemoglobin is an imperfect marker for erythrocytosis as was documented in Tibetan Sherpas who have a normal hemoglobin at high altitude, but increased plasma volume (i.e. *true erythrocytosis* that is *masked* by expanded blood volume). These parameters were measured in Andeans and reported that plasma volume in Andean populations is normal, but their red cell volume is increased (i.e. *true erythrocytosis*). (Stembridge et al., 2019, *PNAS*, 116(33):16177–16179). We have revised and clarified it in the Introduction section of the revised manuscript.

1.2. Is the gene ANP32D related to HIF?

Response: It is known that *ANP32D* is associated with chronic mountain sickness in the different Andean population, Quechua. (Zhou et al., *American Journal of Human Genetics*, 2013, 93(3):452-462). However, whether HIF regulates *ANP32D* has not yet been studied. In our transcriptome analysis, the *ANP32D* gene was not differentially increased in these Aymara Andeans, who exhibit high baseline HIF-transcriptional activity. We now revised the sentence regarding *ANP32D* in Introduction section of our revised manuscript.

1.3. Are the granulocytes RNA representative of the other inflammatory cells in PBMCs?

Response: Granulocytes are an essential component of the innate immune system and are among the first responders to inflammation and infection. As they release pro-inflammatory cytokines, chemokines, and other mediators, granulocytes play a key role in initiating and sustaining the inflammatory process, making them appropriate cells for studying inflammation.

We did not investigate PBMCs for our study because they consist of a mixture of cell types, including lymphocytes (T cells, B cells, NK cells) and monocytes, each with distinct roles in the

immune response. Since gene regulation varies across different cell types, focusing on a single cell type, such as granulocytes, allows for more specific and meaningful insights. Further, the separation of the cells performed in Bolivia, often in challenging environment, generated only pure granulocyte population. We have been unable to obtain sufficiently pure platelets, and another blood cell types, sufficient for rigorous analysis. This limitation is now stated in the revised manuscript.

1.4. While the exon skipping is a novel finding, there is in some figures such as 3, 4, 5, enormous variability and, while statistically significant, i am not sure how significant biologically?

Response: Human samples often exhibit significant individual variation. To reduce this variability, we included as many samples as we could obtain (55 Aymaras and 18 Europeans). While this sample size may not capture all potential differences, we were able to demonstrate statistically significant results and to further support the functional role of *AS-NFKB1* in inflammation and HIF activity, as detailed in our new *in vitro* study presented in the revised manuscript (Figures 6 and 7).

1.5. One of my major issues is that most findings in the paper are correlative to NFKB and HIF. While both are important in the study in what the authors like to prove, there is no proof that is presented by the authors in terms of the importance of NFKB in terms of the adaptation and its importance in regulating in this population the RBC volume which goes to the heart of adaptation.

Response: Thank you for the insightful comment. In the revised manuscript, our new *in vitro* study further demonstrates that *AS-NFKB1* plays a role in the downregulation of both excessive inflammation and HIF activity under inflammatory stress (Figure 7). We plan to explore function of *AS-NFKB1* in erythropoiesis in future research. We have added sentences referring to this in the Discussion section.

Reviewer #2 (Remarks to the Author):

The manuscript “Multi-Omics analysis identifies novel alternative spliced NFKB1 transcripts enriched in Andean Aymara that increase both inflammation and hemoglobin” by Song et al reports that transcript variants of key hypoxia-induced transcription factors may be responsible for population specific high altitude adaptive signatures. The authors further report that these transcript variants may be responsible for observed higher hemoglobin as well as inflammation in Andean Aymara highlanders. This concept is of particular interest to high altitude biology highlighting regulatory roles of alternately spliced transcript variants and observed adaptive features of native highlanders. I have gone through this manuscript with keen interest and my observations are mentioned below:

2.1. The authors have reported transcripts for NFKB1 in Aymara highlanders. However, the (1) protein level of NFKB1 as well as (2) functionality of NFKB1 protein in Aymara highlanders remains elusive in the manuscript. Does the exon skipped transcript variants of NFKB1 results in a functional protein?

Response: Thank you for your comments. To address this important point, we have performed additional experiments. We demonstrated that NFKB1-AS1 and NFKB1-AS3 are not expressed. However, the NFKB1-AS2 (exon5 skipped NFKB1) was expressed, but it is insufficiently transported to the nucleus. In the revised manuscript, these new data are included in the Figure 6.

2.2. The manuscript does not provide any evidence that the exon skipped transcript variants of NFKB1 bind to the downstream targets more efficiently than the canonical NFKB1.

Response: We now added Figure 6 to show that NFKB1/P50 AS2 has comparable inhibitory effect on RelA-dependent promoter activation (e.g., NFKB responsive reporter) compared to the canonical transcript.

2.3. Figure 5 (A, B and C panels) indicate a wide variation of hemoglobin WBC and platelet levels among the studied highlanders. The statistical correlations and significance values have been derived with a handful of samples and the authors believe that this will hold true for a larger cohort of samples. With the variations reported in the manuscript (Hb: 12 – 26 g/dL, WBC: 3 – 9 k/ml and platelets 125 -550 k/ml), it is expected that the statistical correlations will become weak and non-significant in a larger cohort of samples. The authors need to provide experimental evidence for this.

Response: Human samples tend to exhibit considerable individual variation. Since hemoglobin levels differ between males and females, we included only male participants in this manuscript to reduce variation related to the menstrual cycle and pregnancies leading to common iron deficiency in young females that would provide unrelated variable to hemoglobin measurements. We collected samples only from healthy controls with and without CMS. The normal ranges for WBCs and platelets are 4.5-11.0 and 150-450 k/ml, respectively, which is why most of our WBC and platelet counts fall within these ranges. Despite the large variation within the group, the differences between genotypes are statistically significant.

In the new data included in the revised manuscript, we found that the *NFKB1* SNP correlates with inflammatory gene expression levels in polycythemia vera (PV) and essential thrombocythemia (ET), both of which are characterized by high inflammation. In these analyses we now demonstrate that in individuals with baseline increased inflammation to our surprise, the inflammation in these subjects is dampened. This suggests previously unsuspected beneficial effect that may have led to the evolutionary selection of AS-*NFKB1* in Andean Aymaras. We are

now investigating the association of the *NFKB1* SNP with PV and ET disease outcomes. We anticipate that more pronounced differences in hematological parameters (hemoglobin, WBCs, platelets) may be observed in diseases associated with baseline inflammation, see Figure 9S.

2.4. In continuation, the authors need to elucidate the variability of Hb, WBC and platelet levels in the study group reported in Figure 5. It is apparent that the study group comprises of both CMS patients (Hb ≥ 19 g/dL for women and Hb ≥ 21 g/dL for men) and healthy volunteers. A suitable explanation in this aspect is required.

Response: We apologize that we did not clarify this in the manuscript. We collected samples from CMS as well as non-CMS patients. The hemoglobin levels in CMS range from 17.3 to 23.3 g/dL. CMS is not diagnosed based solely on hemoglobin levels. Patients should exhibit symptoms and low oxygen saturation (León-Velarde F, Maggiorini M, Reeves JT, et al. Consensus statement on chronic and subacute high-altitude diseases. *High Alt Med Biol.* 2005;6(2):147-157). Some individuals have hemoglobin levels greater than 21 g/dL but are asymptomatic, indicating that they do not have CMS.

2.5. Lines 397 - 401: The authors have made a statement that Aymara highlanders residing at higher elevations have higher frequency of rs230511-T allele and this genetic adaptation confers better physiological adaptation. According to this statement, residence at higher elevations is directly proportional to higher Hb levels as well as inflammation in Andean Aymara. The authors need to clarify this statement.

Response: We collected only 12 Aymara samples at Chorolque, which was not sufficient to achieve statistical power. For future studies, we would like to collect more Aymara samples from Chorolque. You may be aware that in the high mountain conditions this has been challenging so the trip from La Paz Bolivia to Chorolque at 5616 m takes our collaborators two days using terrain vehicles.

As to higher Hb levels as well as inflammation in Andean Aymara, our original assumption on increased inflammation associated with Aymara evolutionary selected haplotype is now modified. As discussed in preceding replies with baseline increased inflammation, to our surprise, the inflammation in these subjects is dampened; see last paragraph of Result section and Figure 9S.

2.6. Does inflammation is beneficial for high altitude adaptation in Andean Aymara? The authors need to elucidate how increased IL-6 and IFN- γ levels promote or help in promoting high altitude adaptation.

Response: Thank you for your insightful comment. High inflammation may indeed be harmful to the Aymara population. Although IL-6 and IFN- γ levels were increased in *AS-NFKB1*, our new result demonstrated that nuclear NF- κ B protein levels decreased with *AS-NFKB1* overexpression under inflammatory stress induced by TNF treatment, resulting in decreased expressions of inflammatory and HIF-targeted genes (Figure 7). Additionally, we found that the expression levels

of these genes were lower in patients with the rs230511-T variant of *NFKB1* in polycythemia vera (PV) and essential thrombocythemia (ET), both of which are characterized by high baseline inflammatory state (Supplementary figure 9). These findings suggest that although *AS-NFKB1* increases inflammation in the baseline, it reduces inflammatory responses under inflammatory stress compared to *WT-NFKB1*, and the reduced responses may be beneficial at high-altitude at conditions of augmented inflammation.

Reviewer #3 (Remarks to the Author):

Song et al. integrate RNA-seq and whole-genome sequencing data to unravel the molecular signatures of high-altitude adaptation in the Aymara. The authors detect many differentially expressed and differentially spliced genes in granulocytes of Aymara and Europeans. By integrating e/s-QTL mapping and protein-protein interaction networks, they identified *NFKB1* as a candidate hub gene, which also shows novel exon skipping events. Furthermore, Song et al. suggest that the alternative splicing of *NFKB1* plays a role in regulatory changes in inflammatory protein levels, altered regulation of *NFKB1* target genes, including HIFs, and increased Hb levels.

Overall, this is a very interesting study that presents multiple lines of evidence that implicate the alternative splicing of *NFKB1* as a mechanism involved in high-altitude adaptation of Aymara. While I was initially a bit concerned about the relatively small sample size for the initial RNA-seq analyses, the integration of different datasets and validation of the results in a larger number of samples has ultimately convinced me that *NFKB1* might play a role in high-altitude adaptation.

However, the methods section is not very clear in all parts. There seems to be a lot of information missing in the methods section regarding measurements/analyses in the extended dataset. For example, where do the haemoglobin counts come from in Fig.5, and have they been measured in the smaller or the larger dataset, and how have they been measured? In general, it is not always clear which samples were used for which analyses, making it impossible to fully judge these analyses. Thus, the methods section should be extensively reworked and expanded.

Response: Thank you for this comments that led us to improve the manuscript. We apologize for the lack of clarification regarding the samples and methods. In the revised manuscript, we have revised and added more detailed information on sample collection and methodology.

There are many sections throughout the ms that are not well written and don't make sense, making it more difficult than necessary to follow these sections. Please carefully edit the ms.

Minor comments:

- L. 107-108: This sentence doesn't make any sense ("are not found at these genes are not evolutionary selected...").

Response: We made changes in the revised manuscript.

- L. 123-124: This sentence also doesn't make any sense ("might also identify previously undescribed ...")

Response: We made changes in the revised manuscript.

- L.158: Incorrect punctuation/sentence structure: " in (PSI), exon usage..."

Response: We made changes in the revised manuscript.

- The use of the term 'Dysregulation' (e.g. L276) suggests that something is not working correctly and has negative effects. However, that doesn't seem to be the case, as these are adaptations, and regulation has simply changed. Therefore, I would refrain from using the term 'dysregulation' in this context.

Response: We made changes in the revised manuscript.

- L. 312: No information on how Hb levels were measured or what the sample size was for this analysis.

Response: We now add that information in the revised manuscript.

- L. 321: No information on how WBC and platelet counts were obtained and from which samples.

Response: We now add that information in the revised manuscript.

- L. 437-448: How does the choice of cell type potentially influence the results? It would be good to discuss this here as well.

Response: As granulocytes release pro-inflammatory cytokines, chemokines, and other mediators, granulocytes play a key role in initiating and sustaining the inflammatory process, making them suitable cells for studying inflammation. Now we elaborated on this in the Discussion of the revised manuscript.

- L. 503: I am not sure how differential gene expression was determined. In the methods section it says that a Wilcoxon-test was used but in FigS1 it says that DESeq2 was used for identifying DEGs. This should be clarified. Furthermore, I am not sure about using Wilcoxon test is the best approach for such a limited sample size (e.g. see Li et al. 2022 Genome Biology). Which test was chosen and why?

Response: Thank you for providing us with an opportunity to correct it. We agree that Wilcoxon test may be not robust for our analysis while it is good method for analyzing larger sample size. We

identified DEGs by employing DESeq2 which is one of proper approaches addressing a limited sample size. Therefore, we revised the sentence in the Materials and Methods section to that effect.